# Proteasomal degradation of BRAHMA promotes Boron tolerance in *Arabidopsis*

Takuya Sakamoto[1,2], Yayoi Tsujimoto-Inui[1,2], Naoyuki Sotta [2], Takeshi Hirakawa[1], Tomoko M. Matsunaga[3], Yoichiro Fukao[4,5], Sachihiro Matsunaga [1] & Toru Fujiwara[2]

High levels of boron (B) induce DNA double-strand breaks (DSBs) in eukaryotes, including plants. Here we show a molecular pathway of high B-induced DSBs by characterizing *Arabidopsis thaliana hypersensitive to excess boron* mutants. Molecular analysis of the mutants revealed that degradation of a SWItch/Sucrose Non-Fermentable subunit, BRAHMA (BRM), by a 26S proteasome (26SP) with specific subunits is a key process for ameliorating high-B-induced DSBs. We also found that high-B treatment induces histone hyperacetylation, which increases susceptibility to DSBs. BRM binds to acetylated histone residues and opens chromatin. Accordingly, we propose that the 26SP limits chromatin opening by BRM in conjunction with histone hyperacetylation to maintain chromatin stability and avoid DSB formation under high-B conditions. Interestingly, a positive correlation between the extent of histone acetylation and DSB formation is evident in human cultured cells, suggesting that the mechanism of DSB induction is also valid in animals.

[1] Department of Applied Biological Science, Faculty of Science and Technology, Tokyo University of Science, 2641 Yamazaki, Noda, Chiba 278-8510, Japan. [2] Department of Applied Biological Chemistry, Graduate School of Agricultural and Life Sciences, The University of Tokyo, 1-1-1 Yayoi, Bunkyo, Tokyo 113-8657, Japan. [3] Research Institute for Science and Technology, Tokyo University of Science, 2641 Yamazaki, Noda, Chiba 278-8510, Japan. [4] Plant Global Education Project, Nara Institute of Science and Technology, 8916-5 Takayama, Ikoma, Nara 630-0101, Japan. [5] Department of Bioinformatics, Ritsumeikan University, 1-1-1, Nodihigashi, Kusatsu, Shiga 525-8577, Japan. Correspondence and requests for materials should be addressed to S.M. (email: sachi@rs.tus.ac.jp) or to T.F. (email: atoruf@mail.ecc.u-tokyo.ac.jp)

Boron (B) is a trace metalloid element widely required by living organisms including plants and vertebrates[1,2]. In excess, B is toxic to both plants and vertebrates and inhibits their growth and development[1,3,4]. Because of its agricultural importance, the effects of high-B stress on a number of physiological processes in plants have been well described, including aberrations in cell wall development, metabolisms, and cell division and elongation[1,4,5]. In addition, recent studies have identified intranuclear events resulting from high-B stress at the chromatin and DNA levels in eukaryotic cells including animals[6–9]. High doses of B affect the chromatin status by inducing histone hyperacetylation in mouse embryonic cells[6], although whether this occurs in plants is not yet clear. Studies in which *Arabidopsis thaliana*, wheat, and maize were treated with high levels of B revealed that DNA damage is one of the serious adverse effects of high-B stress in plants[7–9]. However, the precise nature of the toxic action of high-B in the nucleus and the molecular mechanisms of tolerance to high-B stress remain poorly understood.

We previously characterized two out of seven *Arabidopsis* mutants isolated as *hypersensitive to excess boron* (*heb*)[7] and found that a chromosomal protein complex, condensin II, is required to alleviate high-B-induced DNA double-strand breaks (DSBs) in root apical meristems (RAMs). The RAM is the base for root growth, consisting of stem cells and their proliferating progenies. Therefore, maintenance of the RAM by repressing DSB accumulation is essential for healthy root growth under high-B conditions[7].

To obtain new insights into the molecular mechanisms underlying tolerance to high-B stress, we characterized three other *heb* mutants, *heb3-1*, *heb6-1*, and *heb6-2*, and found that at least four subunits of the proteasome 19S regulatory particle (RP), RPN2a, RPN8a, RPT2a, and RPT5a, are indispensable for tolerance to high-B stress. Through the analysis of T-DNA insertion mutants, we also found that most of the other subunits are dispensable. RP, together with the 20S core particle (CP), constitutes the 26S proteasome (26SP), a large multienzyme complex that is widely conserved among eukaryotes. The 26SP selectively degrades poly-ubiquitinated (Ub) proteins and is involved in the regulation of numerous processes including cell division, responses to plant hormones, and signaling in response to abiotic and biotic stimuli[10]. The RP acts in the recognition, unfolding, and translocation of targeted proteins for degradation by the CP. Several studies have indicated that individual RP subunits have distinct roles, although some are redundant for specific functions. For instance, RPT2a, RPN10a, and RPN12 are required for the active degradation of a microtubule-associated protein, SPIRAL1, which enables microtubule reassembly for cell elongation under high salinity[11]. Additionally, RPT2a/b and RPT5a directly interact with the transit peptides of plastid precursor proteins for their degradation, but RPT3 and RPN8a do not[12].

Here, we demonstrate that BRAHMA (BRM), a subunit of the SWItch/Sucrose Non-Fermentable (SWI/SNF) chromatin remodeling complex, is a target of a specific composition of the 26SP and that degradation of BRM is crucial for RAM maintenance through alleviation of DSBs under high-B conditions. BRM contains a bromodomain that recognizes acetylated Lys residues on histone tails[13] and is involved in ATP hydrolysis-dependent nucleosome sliding[14]. Besides increasing BRM accumulation, we establish that high-B stress enhances acetylation of histones. Our findings suggest that the 26SP limits the function of BRM in chromatin remodeling in association with histone hyperacetylation to maintain chromatin stability and avoid DSB formation under high-B conditions. Interestingly, histone hyperacetylation

and DSB formation are also observed in human cultured cells treated with high levels of B, indicating B has common molecular actions among eukaryotic cells.

## Results

**Several RP subunits are required for high-B tolerance.** *heb3-1*, *heb6-1*, and *heb7-1* mutants were isolated by screening for mutants that exhibited severe root growth defects under high-B conditions (Fig. 1a, b). Root elongation in these mutants was reduced even under medium-B conditions (0.1–1 mM), whereas root growth in the wild type under these conditions was comparable to that under normal-B conditions (0.03 mM) (Fig. 1b). Although there were significant differences between the wild type and the *heb* mutants, none of the *heb* mutants showed root growth hypersensitivity to other stresses including B-deficiency, arsenite, salinity, and cadmium stresses (Fig. 1c). Oxidative stress is a well-known symptom caused by high-B stress[4]. The root growth of *heb* mutants treated with the reactive oxygen species (ROS)-inducing reagent methyl viologen was no worse than that in the wild type (Supplementary Fig. 1). These results suggested that the *heb* mutations affected the sensitivity to high-B stress specifically and that oxidative stress was not the main cause of their hypersensitivity to high-B stress.

To identify the genes responsible for the *heb* mutants, we used the positional map-based approach with the short-root phenotype as an index. As a result, the locations of the *heb3-1* and *heb7-1* loci were limited to regions of approximately 100 and 24 kb, respectively (Supplementary Fig. 2a, b). Sequence analysis revealed that *heb3-1* and *heb7-1* carried a mutation that resulted in an aberrant ORF in At5g05780 and a nonsense mutation in the predicted fifth exon of At3g05530 (Fig. 2a, b), respectively. Sequence analysis revealed that *heb6-1* also possessed a nonsense mutation in At3g05530, in a distinct position from the mutation in *heb7-1* (Fig. 2b).

At5g05780 and At3g05530 encode *REGULATORY PARTICLE NON-ATPase 8a* (*RPN8a*) and *REGULATORY PARTICLE TRIPLE-A-ATPase 5a* (*RPT5a*), respectively. T-DNA-inserted null mutants of *RPN8a* (*rpn8a-2*)[15] and *RPT5a* (*rpt5a-4*)[16] showed similar short-root phenotypes to those observed in the *heb* mutants under 3 mM B conditions (Fig. 2c, d). Moreover, the introduction of *proRPN8a::gRPN8a-GFP* into *heb3-1* and of *proRPT5a::gRPT5a-GFP* into *heb6-1* and *heb7-1* mutants rescued the hypersensitivity of the *heb* mutants to high-B stress (Fig. 2e, f). The expression of the respective proteins fused with GFP in the transgenic plants was confirmed by confocal laser microscopy (Supplementary Fig. 3). These data established that the causal genes for *heb3* and *heb6, 7* were *RPN8a* and *RPT5a*, respectively. Based on this, we refer to *heb3-1*, *heb6-1*, and *heb7-1* as *rpn8a-3*, *rpt5a-5*, and *rpt5a-6*, respectively, in the present manuscript.

Both RPN8a and RPT5a are subunits of the RP, a subcomplex of the 26SP. The RP is composed of 12 RPN subunits and six RPT subunits. Eight of the RPN and five of the RPT subunits are encoded by paralogous genes in the *Arabidopsis* genome. Distinct subunits or paralogs have redundant functions in gametogenesis[17], organ size regulation[18], and stress responses[19]. It is therefore possible that in addition to RPN8a and RPT5a, other subunits are also essential for tolerance to high-B stress. Among the RP subunit mutants we obtained, two independent alleles of *rpt2a* and *rpn2a* showed hypersensitivity to high-B stress. However, paralogous mutants of the high-B hypersensitive RP subunit mutants, including *rpn8b*, *rpn2b*, *rpt2b-1*, and *rpt5b-3*, were not hypersensitive to high-B stress (Supplementary Fig. 4a–e). RPT6a, RPN3b, and RPN5b are also involved because single alleles of these genes,

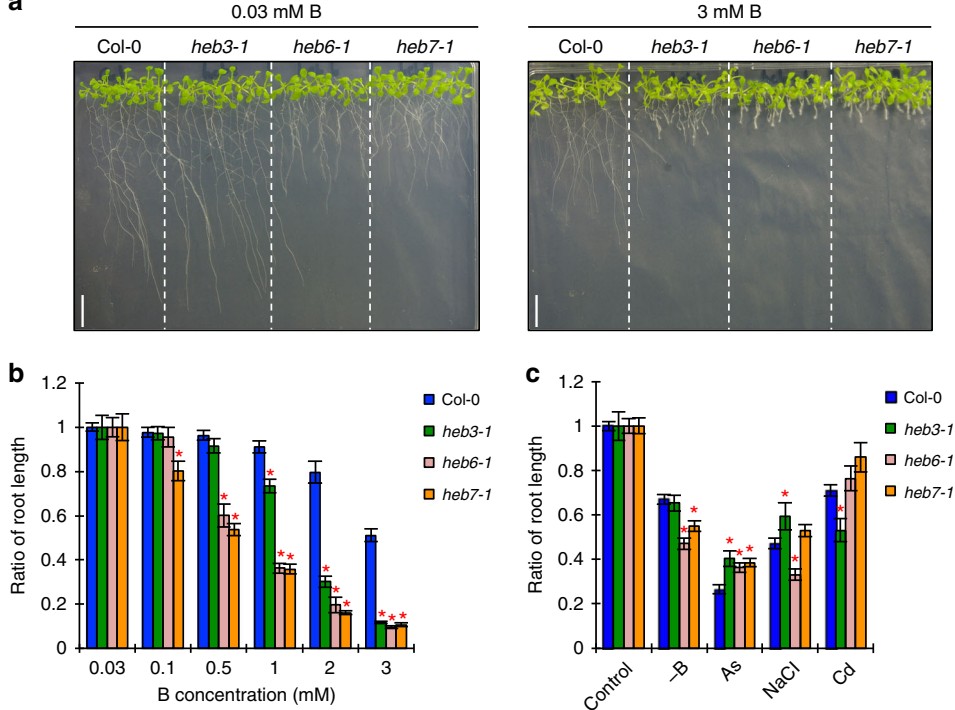

**Fig. 1** The *heb* mutants are specifically hypersensitive to high-B stress. **a** Short-root phenotypes of *heb3-1*, *heb6-1*, and *heb7-1* mutants grown under normal- (left) and high-B (right) conditions for 14 days. Scale bars, 1 cm. **b** Dose-dependent sensitivity of *heb* mutant root growth to high-B stress. Plants were grown on media containing the indicated concentrations of B for 14 days. Values are ratios relative to the value under 0.03 mM B conditions ($n = 13–22$, average ± s.e.m.; *$P < 0.05$, compared with Col-0, Student's *t*-test). **c** Sensitivity of Col-0 and *heb* mutant root growth to various elemental stresses. Plants were grown on media containing no B (−B), 15 μM arsenic (As), 75 mM sodium chloride (NaCl), and 100 μM cadmium (Cd) for 14 days. Values are ratios relative to the value under the control conditions ($n = 13–20$, average ± s.e.m.; *$P < 0.05$, compared with Col-0, Student's *t*-test)

*rpt6a*, *rpn3b-2*, and *rpn5b-3*, showed moderate sensitivity to high-B stress (Supplementary Fig. 4a). It is important to note that not all paralogous subunits are required for the tolerance, suggesting functional differences among paralogs.

To examine whether the effects of high-B stress occur during embryonic and/or post-embryonic stages, we conducted transplant experiments (see details in Methods). In these experiments, high-B stress was applied after the embryonic stages to separate the effects in the embryonic stages from those occurring in post-embryonic stages. The *rpn8a-2* and *rpt5a-4* mutants showed high sensitivities when exposed to high-B conditions (1.5–6 mM) after the embryonic stage, suggesting that RPN8a and RPT5a are involved in high-B tolerance starting at least at post-embryonic stages (Fig. 2e, f and Supplementary Fig. 5). To focus on the post-embryonic stages, we used the transplant method in the experiments described below.

**B accumulation levels in the RP mutants.** Plants can control B uptake through the regulation of B transporters in response to B availability in the soil[20]; therefore, it was possible that the hypersensitivity of the *rpn8a* and *rpt5a* mutants to high-B stress was due to malfunction of B transporters, resulting in damaging B concentrations in the plant body. Under high-B conditions, the B content in the roots of *rpn8a-2* and *rpt5a-4* was less than that in the wild type (Supplementary Fig. 6), suggesting that the RP subunits required for tolerance to high-B stress are not involved in the regulation of B transport systems. In other words, the high-B hypersensitive RP subunit mutants have defects in a cellular mechanism(s) required for tolerance to intracellular toxicity caused by high-B stress.

**Abnormal RAM organization in the high-B treated RP mutants.** Post-embryonic root growth occurs by continuous cell proliferation in the RAM and cell elongation of post-mitotic cells. Previous studies have indicated that high-B stress primarily impairs mitotic activity in the RAM[1,7]. In the present study, we investigated the involvement of specific RP subunits in the maintenance of RAM organization upon high-B stress. Image analysis of propidium iodide (PI)-stained roots by confocal laser microscopy revealed abnormal cell arrangements in the RAMs of *rpt5a* and *rpt2a* mutants exposed to high-B. The number of dead cells, which were recognized by PI staining in the cytoplasm, was increased in the region around the stem cell niche in all mutants upon high-B stress (Fig. 3a and Supplementary Fig. 4f). Measurement of the number of meristematic cortical cells revealed a reduced RAM size in all high-B hypersensitive RP subunit mutants, *rpn8a*, *rpt5a*, *rpt2a*, and *rpn2a*, even under normal conditions (Fig. 3a, b and Supplementary Fig. 4f, g). High-B stress significantly reduced the number of meristematic cortical cells in the wild type and all high-B hypersensitive RP subunit mutants, with greater reductions in the mutants (<28% inhibition in the wild type and >36% inhibition in the mutants) (Fig. 3b and Supplementary Fig. 4g). These results indicated that the RP subunits are responsible for the maintenance of RAM organization especially under high-B conditions. Supporting this idea, the introduction of *proRPN8a::gRPN8a-GFP* into *heb3-1* and of *proRPT5a::gRPT5a-GFP* into *heb6-1* and *heb7-1* mutants at least partially rescued the defects in RAM organization, irrespective of B conditions (Supplementary Fig. 7). The more severe alterations in RAM morphology in the *rpt5a* mutants compared with *rpn8a*/*heb3* mutants exposed to high-B were

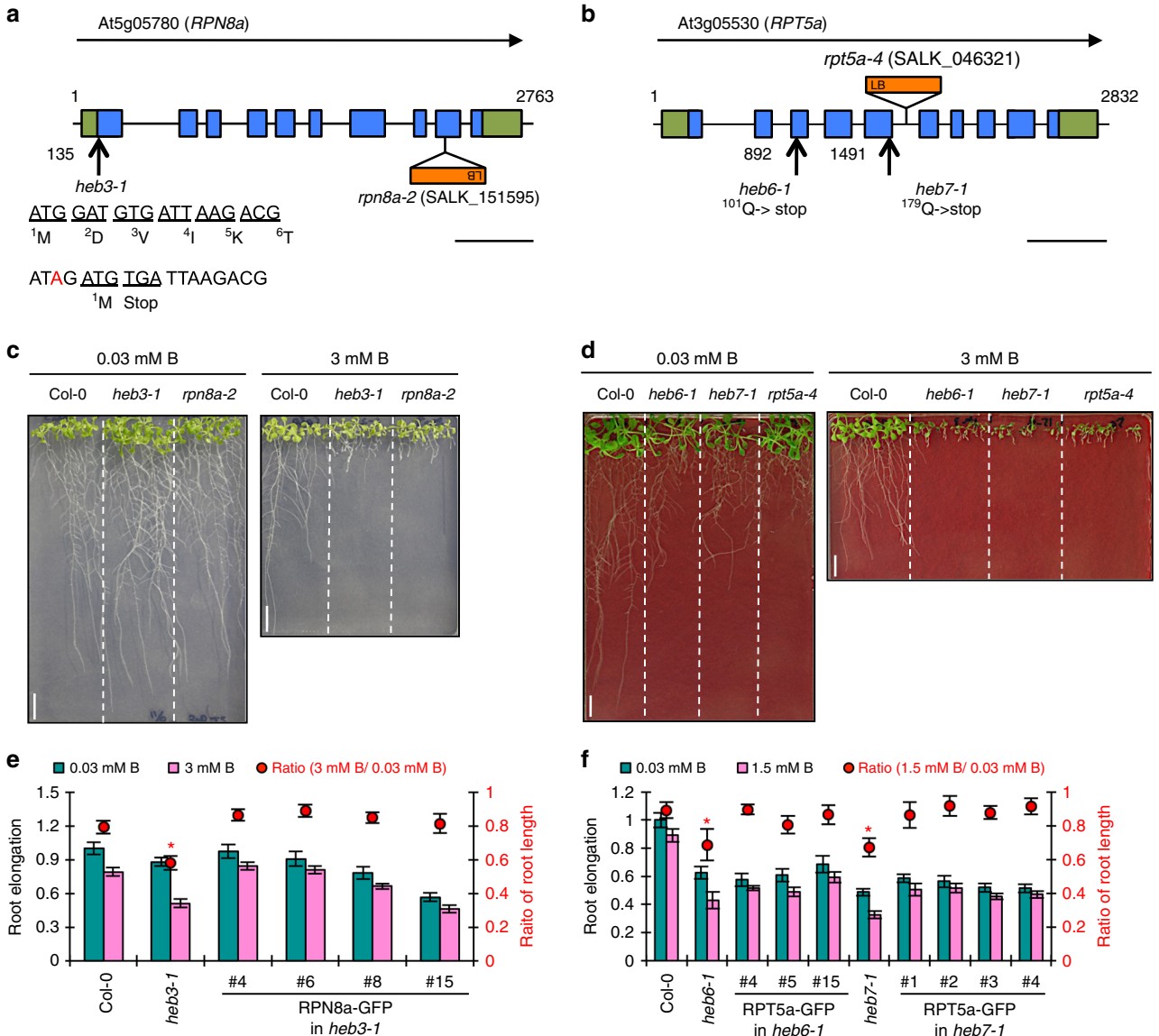

**Fig. 2** *RPN8a* is the gene responsible for *heb3-1* and *RPT5a* is the gene responsible for both *heb6-1* and *heb7-1*. **a**, **b** Gene structure and T-DNA insertion sites in *RPN8a* (**a**) and *RPT5a* (**b**). Blue and green boxes indicate coding and untranslated regions, respectively. Scale bars, 500 bp. **c**, **d** Short-root phenotypes of T-DNA-inserted mutants of *RPN8a* (**c**) and *RPT5a* (**d**) grown under normal- (left) and high-B (right) conditions for 14 days. Scale bars, 1 cm. **e**, **f** Sensitivity of root growth to high-B stress with GFP-fused RPN8a introduced into *heb3-1* (**e**) and GFP-fused RPT5a introduced into *heb6-1* and *heb7-1* mutants (**f**). Sensitivity was assayed by the transplant method (see Methods section). Values for root elongation are ratios relative to the value of Col-0 under 0.03 mM B conditions. Red circles represent the ratio relative to the value under 0.03 mM B conditions ($n = 12$–18, average ± s.e.m.; *$P < 0.05$, compared with Col-0, Student's *t*-test). There were no significant differences in the sensitivity to high-B stress between Col-0 and any of the transgenic plants

consistent with the finding that the *rpt5a* mutants were highly susceptible to high-B stress as indicated by root growth. The *rpt5a* mutants showed the most distinctive phenotypes under high-B conditions; therefore, we continued our evaluation of 26SP function by characterizing *rpt5a* mutants.

**RPT5a functions in ameliorating DSBs in high-B conditions.** We previously established that alleviation of DSBs in the RAM is necessary for tolerance to high-B stress in plants[7]. Even under normal conditions, cell death around the stem cell niche was apparent in the *rpt5a* and *rpt2a* mutants (Fig. 3a and Supplementary Fig. 4f), which is a phenotype consistent with DNA damage[21]. Thus, the RP subunits may act to reduce the extent or

negative effects of DNA damage to improve plant tolerance to high-B stress.

To examine this possibility, we first investigated the sensitivity of *rpt5a* mutants to gamma ray (γ) irradiation and zeocin treatments, which induce transient and constant induction of DSBs, respectively. The root growth of *rpt5a* mutants was more severely inhibited by both γ-irradiation and zeocin treatments than in the wild type (Fig. 4a), indicating that RPT5a functions in the DNA damage response. Next, we assessed DSB accumulation in RAM-containing root tips (less than 1 cm from the tip) of the *rpt5a* mutants by the comet assay and gene expression analysis. As we reported previously[7], the levels of DSB accumulation in root tips and DSB-induced gene expression (*GR1*, *RAD51*, and *BRCA1*) were increased by high-B stress in all plants tested. We

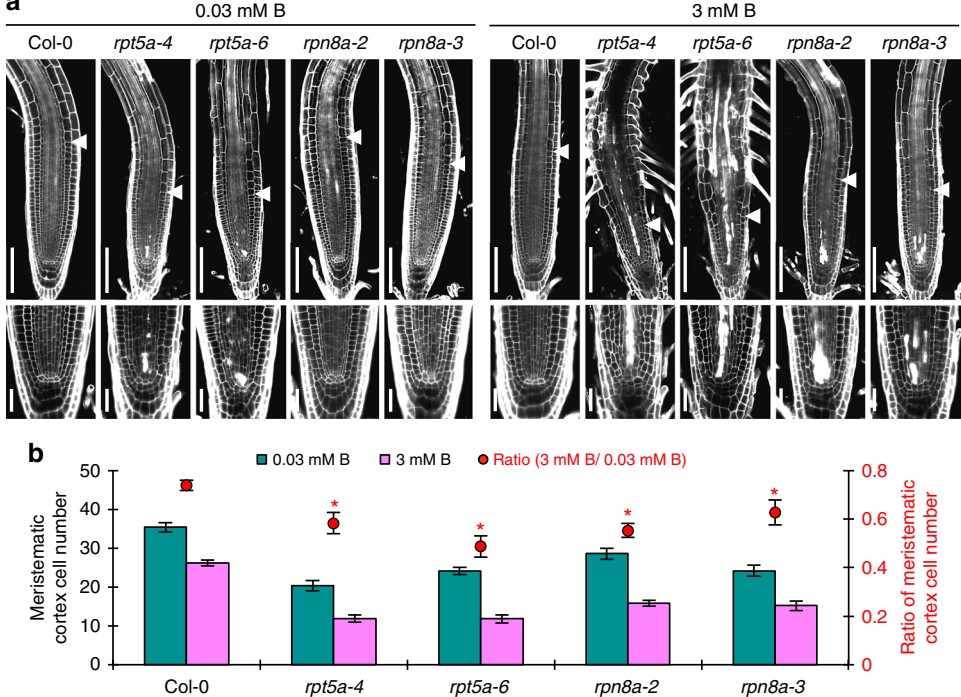

**Fig. 3** Both RPT5a and RPN8a are required for RAM maintenance under high-B conditions. **a** Representative images of Col-0, *rpt5a* mutant, and *rpn8a* mutant root morphology under normal- and high-B conditions. Arrowheads indicate the border between the meristem and elongating region. The bottom panels are magnified images to indicate the area of cell death and cell alignments around the stem cell niche. Scale bars, 50 μm. **b** Effect of high-B stress on the number of cortex cells in the RAM of Col-0, *rpt5a*, and *rpn8a*. Red circles represent the ratio relative to the value under 0.03 mM B conditions (*n* = 10–14, average ± s.e.m.; *P < 0.05, compared with Col-0, Student's *t*-test)

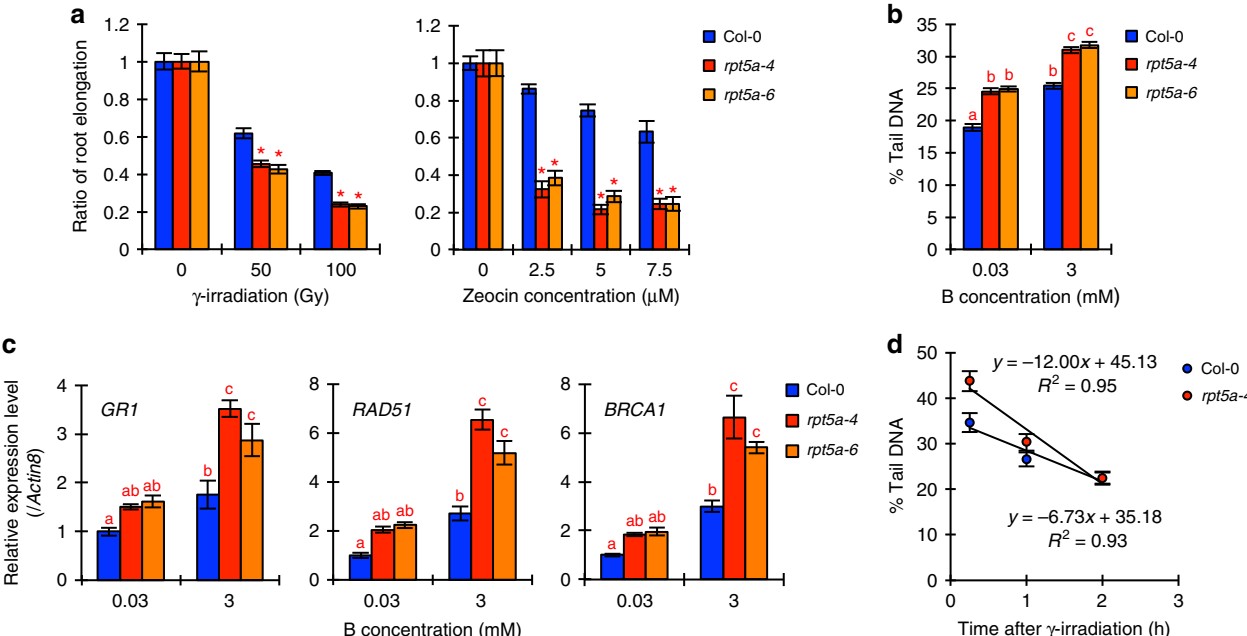

**Fig. 4** RPT5a functions in reducing DSB induction by high-B stress rather than in DSB repair. **a** Sensitivity of root growth to different types of DSB-inducing factors: γ-irradiation and zeocin. Values are ratios relative to the values under each normal condition (*n* = 18–22, average ± s.e.m.; *P < 0.05, compared with Col-0, Student's *t*-test). **b** Levels of DSBs in root tips of Col-0 and *rpt5a* mutant plants grown under normal- and high-B conditions (*n* = 125–132 nuclei, average ± s.e.m.; *P < 0.05, one-way ANOVA and Tukey's HSD). **c** Expression levels of DSB-inducible genes in root tips of Col-0 and *rpt5a* mutant plants grown under normal- and high-B conditions (*n* = 4, average ± s.e.m.; *P < 0.05, one-way ANOVA and Tukey's HSD). **d** DSB repair rate in Col-0 and *rpt5a-4* root tips after transient DSB induction by γ-irradiation, represented using data from Supplementary Fig. 8

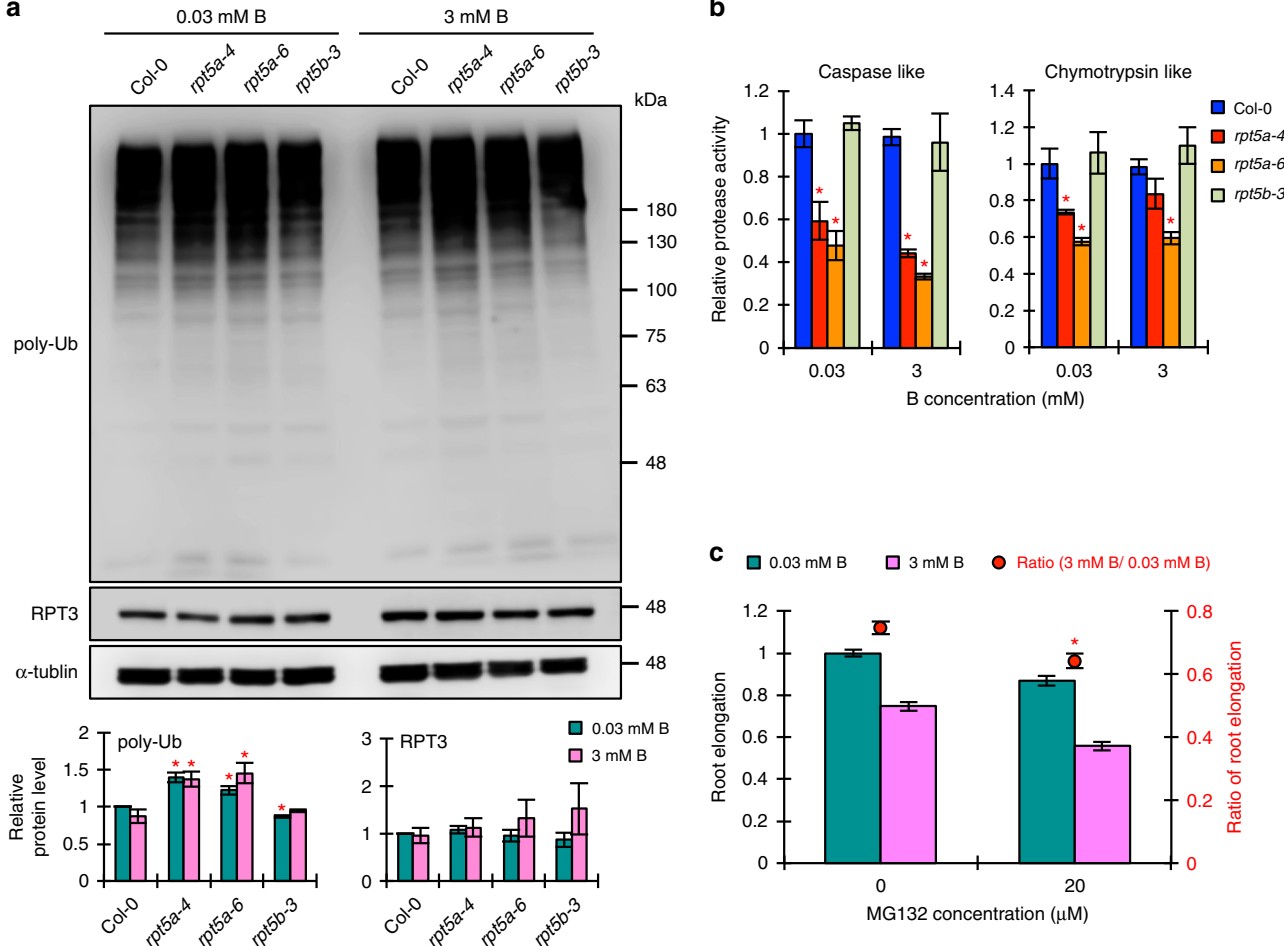

**Fig. 5** Impaired 26SP activity causes the hypersensitivity to high-B stress. **a** Accumulation profile of poly-Ub proteins and amount of 26SP (represented by the amount of RPT3) in roots of Col-0 and *rpt5* mutant plants grown under normal- and high-B conditions. Lower panels show the relative level of indicated protein accumulation normalized against α-tubulin (*n* = 3–4, average ± s.e.m.; *P < 0.05, compared with Col-0 under 0.03 mM B conditions, Student's *t*-test). **b** Peptidase activities of 20SP in Col-0 and *rpt5* mutant roots grown under normal- and high-B conditions (*n* = 4, average ± s.e.m.; *P < 0.05, compared with Col-0, Student's *t*-test). **c** Effect of MG132 treatment on the sensitivity of Col-0 root growth to high-B stress. Values for root elongation are ratios relative to the value under normal conditions. Red circles represent the ratio relative to the value under 0.03 mM B conditions (*n* = 25–27, average ± s.e.m.; *P < 0.05, compared with 0 µM MG132 conditions, Student's *t*-test)

found that both levels were higher in the *rpt5a* mutants than in the wild type, irrespective of B conditions (Fig. 4b, c). These results suggested that the 26SP containing RPT5a functions in ameliorating DNA damage, which is required for the tolerance to high-B stress.

There are two fundamental mechanisms that alleviate DNA damage: maintenance of high chromatin stability to avoid damage and prompt repair of damaged DNA. To understand the point of action of RPT5a, its activity in the repair of damaged DNA was examined. We calculated the repair kinetics of DSBs in transiently γ-irradiated root tips and expressed the kinetics as the rate of DSB reduction. Although the initial increases in DSB levels after γ-irradiation were similar between the wild type and *rpt5a-4* (an approximately 20% increase in % Tail DNA), DSB levels were more reduced in *rpt5a-4* than in the wild type as time advanced (Supplementary Fig. 8). At 2 h after γ-irradiation, the DSB levels in *rpt5a-4* were comparable to the background level (Supplementary Fig. 8); therefore, the repair kinetics were evaluated by least-square linear regression of fractions of the DSB level representing the remaining induced-DSBs at 0.25, 1, and 2 h after γ-irradiation. The % Tail DNA value decreased at a higher rate in *rpt5a-4* than in the wild type (Fig. 4d), suggesting that DNA

repair activity was enhanced in *rpt5a-4*. Considering that DSB levels were higher in the *rpt5a* mutants than in the wild type (Fig. 4b), even under normal conditions, the 26SP containing RPT5a likely functions in the maintenance of high chromatin stability to prevent DNA damage occurrence rather than in repair of damaged DNA.

**Total 26SP activity is required for high-B tolerance.** The RP functions in the recognition and unfolding of poly-ubiquitin tagged 26SP substrates. Defects in RP subunits, such as RPT2a, RPN10, and RPN12a, affect the ubiquitin-dependent proteolytic activity of the 26SP[22]. Therefore, mutations in the high-B hypersensitive RP subunits could also affect 26SP activity. Our immunoblot analysis showed increased accumulation of poly-Ub proteins in the roots of *rpt5a* but not *rpt5b-3* mutants compared to the wild type roots, irrespective of B conditions (Fig. 5a). The intensity of the RPT3 bands indicated that the amount of 26SP was similar among the tested plants and not affected by the high-B treatment (Fig. 5a). 26SP proteolytic activities, which were assessed by measuring caspase-like and chymotrypsin-like activities, were reduced in the roots of *rpt5a* but not *rpt5b-3* mutants, irrespective of B conditions (Fig. 5b). These results indicated that

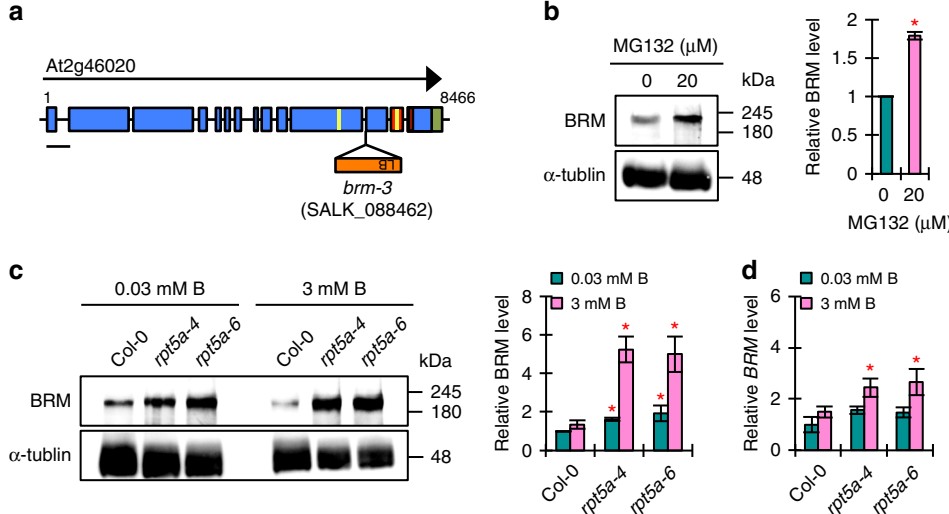

**Fig. 6** A 26SP containing RPT5a regulates BRM levels in roots. **a** Gene structure of *BRM*. Blue and green boxes indicate coding regions and untranslated regions, respectively. Red and yellow bars indicate regions encoding PEST motifs and KEN-boxes, respectively. Scale bar, 500 bp. **b** Effect of MG132 treatment on BRM accumulation in Col-0 roots. The right panel shows the relative level of BRM accumulation normalized against α-tubulin ($n = 3$, average ± s.e.m.; $P < 0.05$, Student's *t*-test). **c** BRM levels in the roots of Col-0 and *rpt5a* mutant plants grown under normal- and high-B conditions. The right panel shows the relative level of BRM accumulation normalized against α-tubulin ($n = 3$, average ± s.e.m.; $P < 0.05$, compared with Col-0 under 0.03 mM B conditions, Student's *t*-test). **d** *BRM* mRNA levels in the root tips of Col-0 and *rpt5a* mutant plants grown under normal- and high-B conditions ($n = 4$, average ± s.e.m.; $P < 0.05$, compared with Col-0 under 0.03 mM B conditions, Student's *t*-test)

the degree of ubiquitin-dependent 26SP proteolytic activity correlates with plant sensitivity to high-B stress. Furthermore, treatment with MG132, an inhibitor of 26SP protease activities, enhanced the sensitivity of wild type roots to high-B stress (Fig. 5c). The ratio of root elongation between the control and high-B conditions was further decreased by MG132 treatment. These results strongly suggested that 26SP proteolytic activity is a requirement for the high-B tolerance.

**Specifically accumulated proteins in high-B-treated *rpt5a*.** The finding that the level of overall 26SP activity was related to high-B hypersensitivity suggested the existence of target proteins to be degraded under high-B conditions. Considering their impaired overall 26SP activity, the *rpt5a* mutants should show specific accumulation of such target proteins. To identify these proteins, we conducted proteomic analysis of poly-Ub proteins in roots. Poly-Ub proteins were purified and enriched from the roots of wild type and *rpt5a* seedlings grown under normal- and high-B conditions using resin conjugated with GST-hHR23B-UBA that selectively recognized the poly-Ub moiety on proteins but not free Ub. Enrichment of poly-Ub proteins by purification was confirmed by immunoblot analysis (Supplementary Fig. 9a). The purified proteins were subjected to LC-MS/MS analysis (Supplementary Fig. 9b). More than 64 proteins were identified from all plants tested (Supplementary Fig. 9c and Supplementary Data 1) and comparative analysis revealed 12 proteins that were specifically accumulated in the *rpt5a* mutants treated with high-B (Supplementary Table 1 and Supplementary Fig. 9d). Potential poly-ubiquitination of the identified proteins was predicted by analyzing ubiquitination-targeting motifs—PEST motifs, KEN-boxes, and D-boxes. Eleven of the identified proteins contained at least one of these motifs, suggesting that these proteins are potential substrates of ubiquitin-dependent proteolysis via the 26SP.

**The 26SP containing RPT5a regulates BRM levels in roots.** As our results implied the involvement of the 26SP containing RPT5a in maintaining chromatin stability, we focused on BRM

among the potential substrates. BRM is a member of the SWI2/SNF2 subgroup of chromatin remodeling ATPases[23] and contains two putative PEST motifs and one putative KEN-box (Supplementary Table 1 and Fig. 6a). Immunoblot analysis revealed that BRM accumulation was increased by MG132 treatment (Fig. 6b). Moreover, we confirmed that BRM protein accumulation was higher in the *rpt5a* mutants than in the wild type (Fig. 6c). These results established that BRM is a target of the 26SP containing RPT5a. We also found that BRM accumulation was further promoted by high-B stress only in the *rpt5a* mutants (Fig. 6c). Interestingly, the levels of *BRM* mRNA in the mutants and following high-B stress mirrored those of the BRM protein (Fig. 6d). However, the extents of the differences in mRNA accumulation between the wild type and the *rpt5a* mutants were much less than those observed in protein accumulation under high-B conditions (Fig. 6c, d).

**BRM is a determinant of high-B hypersensitivity in *rpt5a*.** We next examined whether BRM plays a crucial role in adaptation to high-B stress. Transgenic plants showing enhanced BRM expression were generated using a constitutive promoter (Supplementary Fig. 10a). Four out of eight transgenic lines exhibited poorer root growth under high-B conditions but not under normal conditions than the wild type (Fig. 7a). In addition, two representative transgenic lines showed higher reduction ratios for RAM size under high-B stress conditions and increased DSB levels only under high-B stress conditions (Supplementary Fig. 10b–d). These results suggested that BRM accumulation has an inhibitory effect on root elongation especially under high-B conditions. Furthermore, the extent of root growth inhibition by high-B stress was lower in *brm-3*, a T-DNA insertion mutant of BRM (Fig. 6a), than in the wild type (Fig. 7b). Moreover, the RAM size in *brm-3* was relatively well maintained compared with that in the wild type under all high-B conditions (Fig. 7c, d). These results established that *brm-3* had high tolerance to high-B stress. In other words, BRM is a negative factor for *Arabidopsis* root growth under high-B stress.

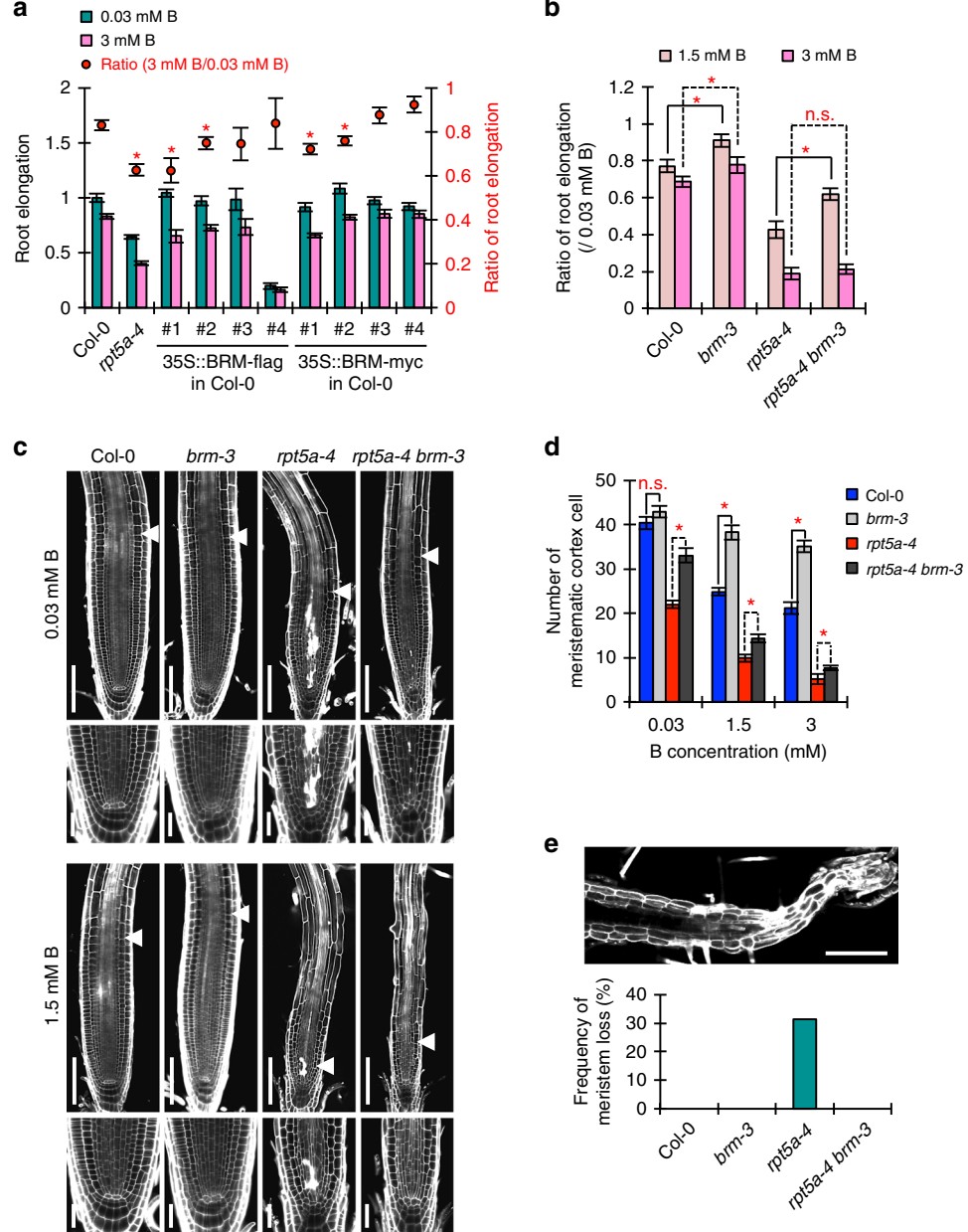

**Fig. 7** Regulation of BRM levels through the 26SP containing RPT5a is crucial for RAM maintenance under high-B conditions. **a** Sensitivity of root growth of Col-0, *rpt5a-4*, and transgenic plants showing enhanced *BRM* expression to high-B stress. Values for root elongation are ratios relative to the value of Col-0 under 0.03 mM B conditions. Red circles represent the ratio relative to the value under 0.03 mM B conditions ($n = 10$–$17$, average ± s.e.m.; *$P < 0.05$, compared with Col-0, Student's *t*-test). **b** Sensitivity of Col-0, *brm-3*, *rpt5a-4*, and *rpt5a-4 brm-3* root growth to high-B stress. Values are ratios relative to the values at 0.03 mM B ($n = 15$–$16$, average ± s.e.m.; *$P < 0.05$, Student's *t*-test). **c** Representative images of root morphology of Col-0, *brm-3*, *rpt5a-4*, and *rpt5a-4 brm-3* treated with normal- and high-B for 7 days. Arrowheads indicate the border between the meristem and elongating region. Below are magnified images of the stem cell niche. Scale bars, 50 μm. **d** Effect of high-B stress on the number of cortex cells in the RAM of Col-0, *brm-3*, *rpt5a-4*, and *rpt5a-4 brm-3* ($n = 15$–$22$, average ± s.e.m.; *$P < 0.05$, Student's *t*-test). **e** Meristem loss specifically observed in *rpt5a-4* with long-term (7 days) treatment of 3 mM B. Upper panel shows a representative image of a *rpt5a-4* root with meristem loss. Lower panel shows the frequency of meristem loss in Col-0, *brm-3*, *rpt5a-4*, and *rpt5a-4 brm-3* under 3 mM B conditions, which were evaluated using same data sets used in **d** ($n = 15$–$22$)

To examine whether BRM overaccumulation was one of the main causes of the *rpt5a* hypersensitivity to high-B stress, we crossed *rpt5a-4* with *brm-3*. The double mutant, *rpt5a-4 brm-3*, showed much less sensitivity to a moderate level of B stress (1.5 mM) than did *rpt5a-4* in terms of root growth, cell death frequency, RAM structure and RAM size, but was still more sensitive than the wild type (Fig. 7b–d). However, under a high level of B stress (3 mM), while *rpt5a-4 brm-3* did not show increased root growth tolerance (Fig. 7b), the RAM size in *rpt5a-4*

was rescued by the additional BRM defect (Fig. 7c, d). Moreover, *rpt5a-4 brm-3* did not have RAM-deficient roots, which appeared when *rpt5a-4* was treated for a long term (7 days) with high-B (Fig. 7e). Taken together, these observations indicated that the 26SP containing RPT5a is involved in the regulation of BRM levels in response to B conditions, which is crucial for suppressing the severe defects in RAM organization and subsequent root growth, especially under high-B conditions. It should be noted that the root growth of *rpt5a-4* was not fully rescued by the *brm-3*

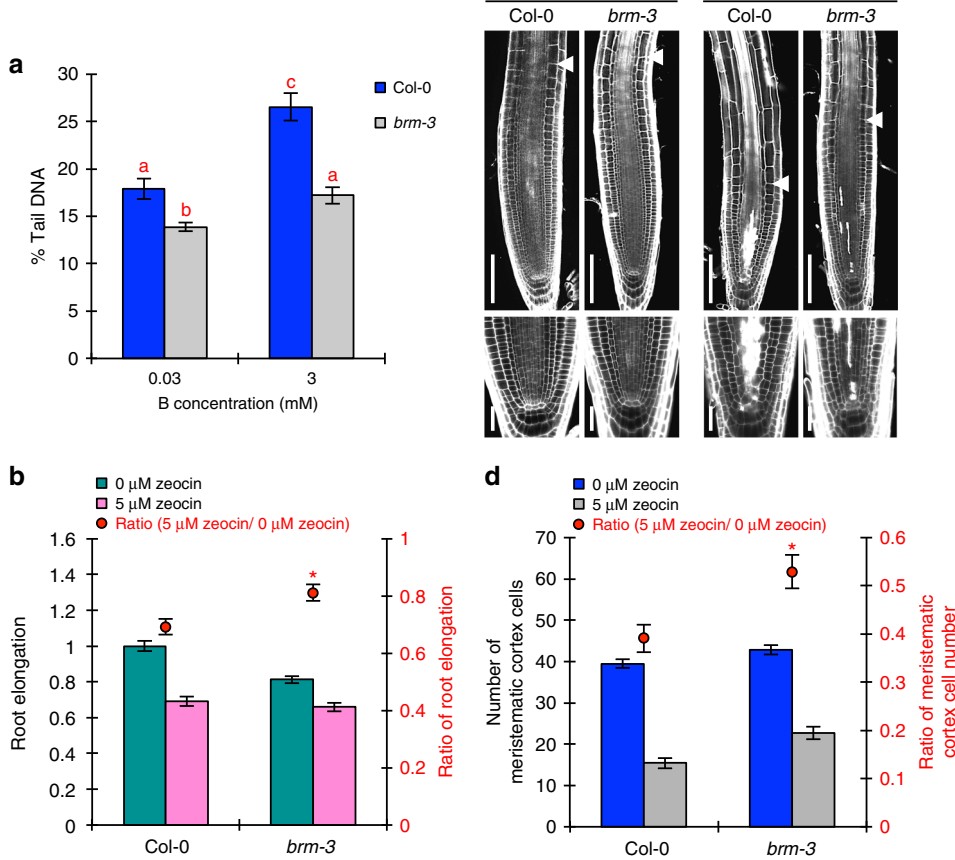

**Fig. 8** Impairment of BRM function confers DNA-damage tolerance in roots. **a** Levels of DSBs in root tips of Col-0 and *brm-3* grown under normal- and high-B conditions ($n = 111–115$, average ± s.e.m.; $P < 0.05$, one-way ANOVA and Tukey's HSD). **b** Sensitivity of Col-0 and *brm-3* root growth to the DSB-inducing factor zeocin. Values for root elongation are ratios relative to the value of Col-0 under 0 μM zeocin conditions. Red circles represent the ratio relative to the value under 0 μM zeocin conditions ($n = 19–20$, average ± s.e.m.; *$P < 0.05$, compared with Col-0, Student's *t*-test). **c** Representative images of root morphology of Col-0 and *brm-3* treated with zeocin. Arrowheads indicate the border between the meristem and elongating region. Below are magnified images of the stem cell niche. Scale bars, 50 μm. **d** Effect of zeocin treatment on the number of cortex cells in the RAM of Col-0 and *brm-3* ($n = 20–24$, average ± s.e.m.; *$P < 0.05$, compared with Col-0, Student's *t*-test)

mutation under high-B conditions (Fig. 7b), suggesting that the overaccumulation of BRM is not the sole cause of the hypersensitivity of *rpt5a* mutants to high-B stress.

**BRM has adverse effects on amelioration of DSBs in the RAM**. Given that DNA damage is the main defect caused by high-B stress, it is possible that the high tolerance of *brm-3* to high-B stress may be conferred by the amelioration of DSBs. Consistent with this idea, we found that DSB accumulation through high-B stress in root tips was suppressed in *brm-3* (Fig. 8a). To elucidate whether DNA damage amelioration in *brm-3* was specific to high-B-induced DSBs, we investigated the sensitivity of *brm-3* to zeocin-induced DSBs. The inhibition of root growth (Fig. 8b) and the reduction in RAM size (Fig. 8c, d) by zeocin treatment were less pronounced in *brm-3* than in the wild type. Moreover, cell death in the stem cell niche caused by zeocin-induced DSBs was suppressed in *brm-3* (Fig. 8c). These results indicated that BRM has a negative effect on the suppression of DSBs caused by various DNA-damaging factors, including high-B stress. The chromatin remodeling complex containing BRM acts in chromatin opening[14]. Therefore, the high tolerance of *brm-3* to DNA-damaging factors may be caused by an increase in the number of closed chromatin regions.

However, the enhanced expression of BRM alone did not cause increased DSB levels under normal conditions or increased sensitivity to zeocin-induced DSBs (Supplementary Fig. 10d, e). These results imply that the negative effect of BRM on the suppression of DSBs involves additional factors and that the molecular action leading to DSBs differs between high-B and zeocin.

**High-B stress induces histone hyperacetylation**. BRM needs to bind to acetylated Lys residues on histone tails through its bromodomain to function normally[24]. The accumulation of BRM in response to high-B stress led us to speculate that histone hyperacetylation occurs under high-B conditions. It is conceivable that a high level of intra-nuclear B itself (as boric acid or borate ions) directly or indirectly reduces histone deacetylase (HDAC) activity in plant roots. It has been demonstrated previously that boric acid inhibits the activity of HDACs in vitro, and treatment with excess boric acid results in histone hyperacetylation in mouse embryos[6], supporting our speculation. In this study, we found that the acetylation level of histone H3 was increased by high-B treatment in the wild type (Fig. 9a). In addition, we conducted a competition experiment between HDAC inhibitor, trichostatin A (TSA), and boric acid. TSA application alone suppressed root elongation

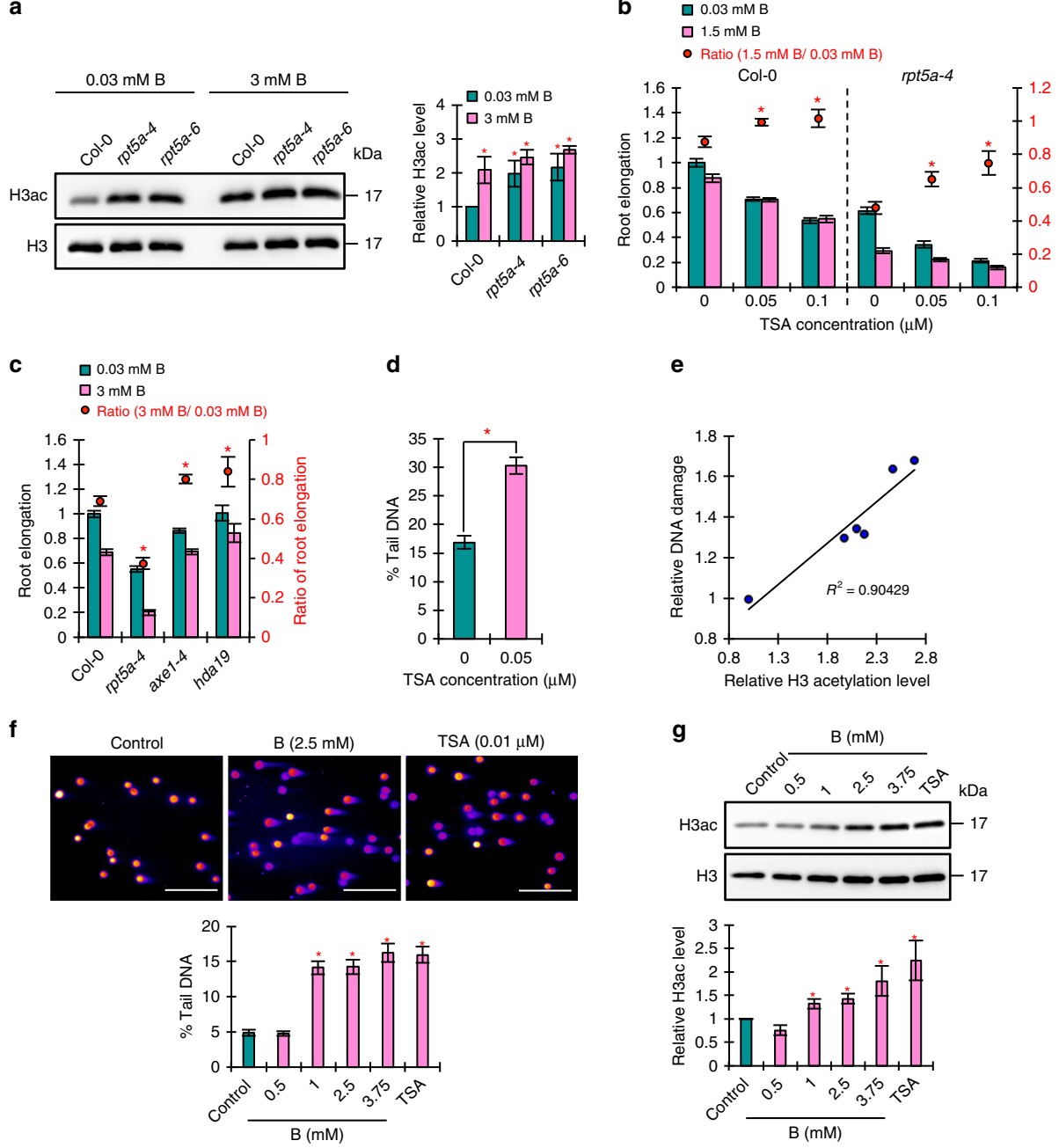

**Fig. 9** Histone hyperacetylation is the main cause of DSB induction under high-B stress. **a** Levels of histone H3 acetylation in the roots of Col-0 and *rpt5a* mutant plants grown under normal- and high-B conditions. The right panel shows the relative level of acetylated histone H3 accumulation normalized against total histone H3 ($n = 3$, average ± s.e.m.; *$P < 0.05$, compared with Col-0 under 0.03 mM B conditions, Student's *t*-test). **b** Effect of histone hyperacetylation induced by TSA treatment on the sensitivity of Col-0 and *rpt5a-4* to high-B stress. Values for root elongation are ratios relative to the value of Col-0 under 0 μM TSA conditions. Red circles represent the ratio relative to the value under 0.03 mM B conditions without TSA ($n = 17$–19, average ± s.e.m.; *$P < 0.05$, compared with 0 μM TSA conditions, Student's *t*-test). **c** Sensitivity of root growth of Col-0 and *hdac* mutants to high-B stress. Values for root elongation are ratios relative to the value of Col-0 under 0.03 mM B conditions. Red circles represent the ratio relative to the value under 0.03 mM B conditions ($n = 15$–20, average ± s.e.m.; *$P < 0.05$, compared with Col-0, Student's *t*-test). **d** Effect of histone hyperacetylation on levels of DSBs in Col-0 root tips ($n = 102$–111, average ± s.e.m.; *$P < 0.05$, compared with 0 μM TSA conditions, Student's *t*-test). **e** Relationship between levels of histone H3 acetylation and DSB accumulation under high-B conditions according to data from Col-0, *rpt5a* mutants, *brm-3*, and two BRM high-expression transgenic plants. **f** Representative images of comet assays with high-B- and TSA-treated HeLa cells. Scale bars, 50 μm. The lower panel shows the levels of DSBs in HeLa cells treated with various levels of B and 0.01 μM TSA ($n = 105$, average ± s.e.m.; *$P < 0.05$, compared with control conditions, Student's *t*-test). **g** Levels of histone H3 acetylation in HeLa cells treated with various levels of B and 0.01 μM TSA. The lower panel shows the relative level of acetylated histone H3 accumulation normalized against total histone H3 TSA ($n = 4$–5, average ± s.e.m.; *$P < 0.05$, compared with control conditions, Student's *t*-test)

(Fig. 9b). Simultaneous treatment of B and TSA resulted in higher ratios of root elongation in both the wild type and *rpt5a-4* compared with B treatment alone (Fig. 9b), indicating that the actions of TSA and high-B stress are competitive. Consistent with this, mutations in *HDAC6* (*axe1-4*) and *HDAC19* (*hda19*), which lead to increased genome-wide histone acetylation[25,26], reduced sensitivity to high-B stress (Fig. 9c). Overall, these data demonstrated that high-B stress induces histone hyperacetylation, likely by inhibiting HDAC activity.

We also analyzed the effect of high-B stress on other histone modifications, H3K9me1 and H3K4me3, which are related to closed and open chromatin, respectively[27,28]. Total levels of H3K9me1 were decreased by high-B stress in both the wild type and *rpt5a* mutants, which suggested increased chromatin opening. In contrast, H3K4me3 levels did not change after high-B stress in either the wild type or the *rpt5a* mutants (Supplementary Fig. 11). It should be noted that even though the total level did not change, there could have been a change in the distribution across the genome. Taken together, these results indicated that high-B stress also affects other histone modifications. However, it is unknown whether the change in other modifications is a consequence of histone hyperacetylation caused by high-B stress or the direct effects of high-B stress.

**Histone hyperacetylation is linked to DSB induction**. Histone hyperacetylation is a well-known cause of interphase chromatin decondensation[29], which results in enhanced chromatin vulnerability to DNA-damaging factors[30]. We confirmed that TSA treatment reduced the tolerance of chromatin to micrococcal nuclease (MNase) (Supplementary Fig. 12a), indicating that histone hyperacetylation induces chromatin sensitivity to DNA-damaging factors in *Arabidopsis*. We also found that DSBs were highly accumulated after TSA treatment (Fig. 9d). Thus, we next investigated whether histone hyperacetylation was linked to the extent of DSB formation under high-B conditions. In addition to enhancing histone acetylation (Fig. 9a), high-B treatment reduced the tolerance of chromatin to MNase in the wild type (Supplementary Fig. 12b). Compared with the wild type, histone acetylation levels were increased and chromatin stability was reduced in the *rpt5a* mutants grown under normal conditions and these phenotypes were enhanced under high-B conditions (Fig. 9a and Supplementary Fig. 12b). Taking the higher DSB accumulation in the *rpt5a* mutants irrespective of B stress (Fig. 4b) into account, DSB levels were positively correlated with the histone acetylation level in roots (Fig. 9e). Thus, the level of DSB formation can be attributed to histone hyperacetylation under high-B conditions in roots. Taken together, these data demonstrated that high-B stress promotes chromatin destabilization caused by histone hyperacetylation, which results in increased susceptibility to DNA-damaging factors.

The genotoxicity of B in mammalian cells is not well characterized, although high-B dependent histone hyperacetylation has been reported[6]. We evaluated the effect of high levels of B on human cultured cells. HeLa cell proliferation is suppressed under B concentrations more than 1 mM[31]. We found that treatment with more than 1 mM B caused DSBs in HeLa cells (Fig. 9f) and promoted histone acetylation (Fig. 9g), similar to the case in plant cells. In addition, TSA treatment also induced DSBs in HeLa cells and the total DSB levels were consistent with histone acetylation levels (Fig. 9f, g), again similar to the case in plant cells. These data suggested that high-B-induced DSBs also occur in mammalian cells, and result from increased levels of histone acetylation, as observed in plant cells.

**BRM enhances the effects of histone hyperacetylation**. Considering that the acetylation levels of histone H3 were comparable between the wild type and the *rpt5a* mutants under high-B conditions (Fig. 9a), histone hyperacetylation alone does not determine the sensitivity to high-B stress. Thus, we hypothesized that some factor that exacerbates the negative effects of histone hyperacetylation could be a cause of the high-B hypersensitivity in the *rpt5a* mutants. First, we investigated the sensitivity of the *rpt5a* mutants to histone hyperacetylation-inducing chemicals. We found that root growth in the *rpt5a* mutants was highly sensitive to HDAC inhibitors, TSA and sodium butyrate (NaBT) (Fig. 10a), similarly to high-B stress (3 mM B) (Supplementary Fig. 5), indicating that histone hyperacetylation is a key process for evoking high-B hypersensitivity in the *rpt5a* mutants. Next, we focused on the relationship between BRM and histone hyperacetylation. Analysis of published chromatin immunoprecipitation-seq (ChIP-seq) data related to BRM, H3K9ac, H3K14ac, and H3K27me3[32,33] revealed that the BRM was highly enriched around sites of H3K9ac and H3K14ac, but not of H3K27me3, suggesting the possible binding of BRM to acetylated histone residues (Fig. 10b). Then, we found that *brm-3* was tolerant to TSA treatment and, conversely, enhanced BRM expression caused high sensitivity to TSA treatment (Fig. 10c, d). However, BRM levels did not affect the levels of histone acetylation (Fig. 10e). These data indicated that the negative effects of histone hyperacetylation on genomic stability are at least in part dependent on the levels of BRM.

Overall, our results suggested that the hypersensitivity of the *rpt5a* mutants to high-B stress is attributable mainly to the high accumulation of BRM, which enhances the negative effects of histone hyperacetylation under high-B stress conditions. It is conceivable that the distinct responses of enhanced BRM expressing lines to zeocin and high-B treatments (Supplementary Fig. 10e) is attributed to their capability of inducing histone hyperacetylation. In fact, zeocin is known to be a radiomimetic drug that can cause oxidative damage in DNA through producing ROS[21].

**Discussion**

Our analyses of the *rpt5a* mutants confirmed that DNA damage in the RAM is a major cause of the root growth inhibition induced by high-B stress in *Arabidopsis* (Supplementary Fig. 13), as described in our previous study[7]. Moreover, we established that high-B stress induces histone hyperacetylation in plant roots. The level of histone hyperacetylation is correlated with the level of DSB accumulation in the RAM (Fig. 9e). Thus, we suggest that B-dependent histone acetylation is an underlying cause of high-B-induced DNA damage. Histone acetylation is a major factor that affects the degree of chromatin condensation. Acetylation neutralizes the positive charges on Lys residues in histone tails, preventing the interaction between histones and negatively charged DNA, leading to opening and loosening of the chromatin configuration[34]. Loosened chromatin is susceptible to DNA-damaging factors[30]. We found that plants showing elevated histone acetylation levels were hypersensitive to exogenous DSB-inducing factors (Figs. 4a and 9a). In addition to exogenous DNA-damaging factors, such as heavy metals, chemicals, and cosmic rays, DNA damage is constantly caused by endogenous factors, including hydrolysis and exposure to ROS and other reactive metabolites[35,36]. Although we cannot completely exclude the possibility that the direct binding of B to DNA is a cause of DSBs, considering the chemical properties of borate—it can bind to ribose in ribonucleotides and RNA but does not bind to DNA —this is unlikely[37]. Therefore, we propose that the high-B-

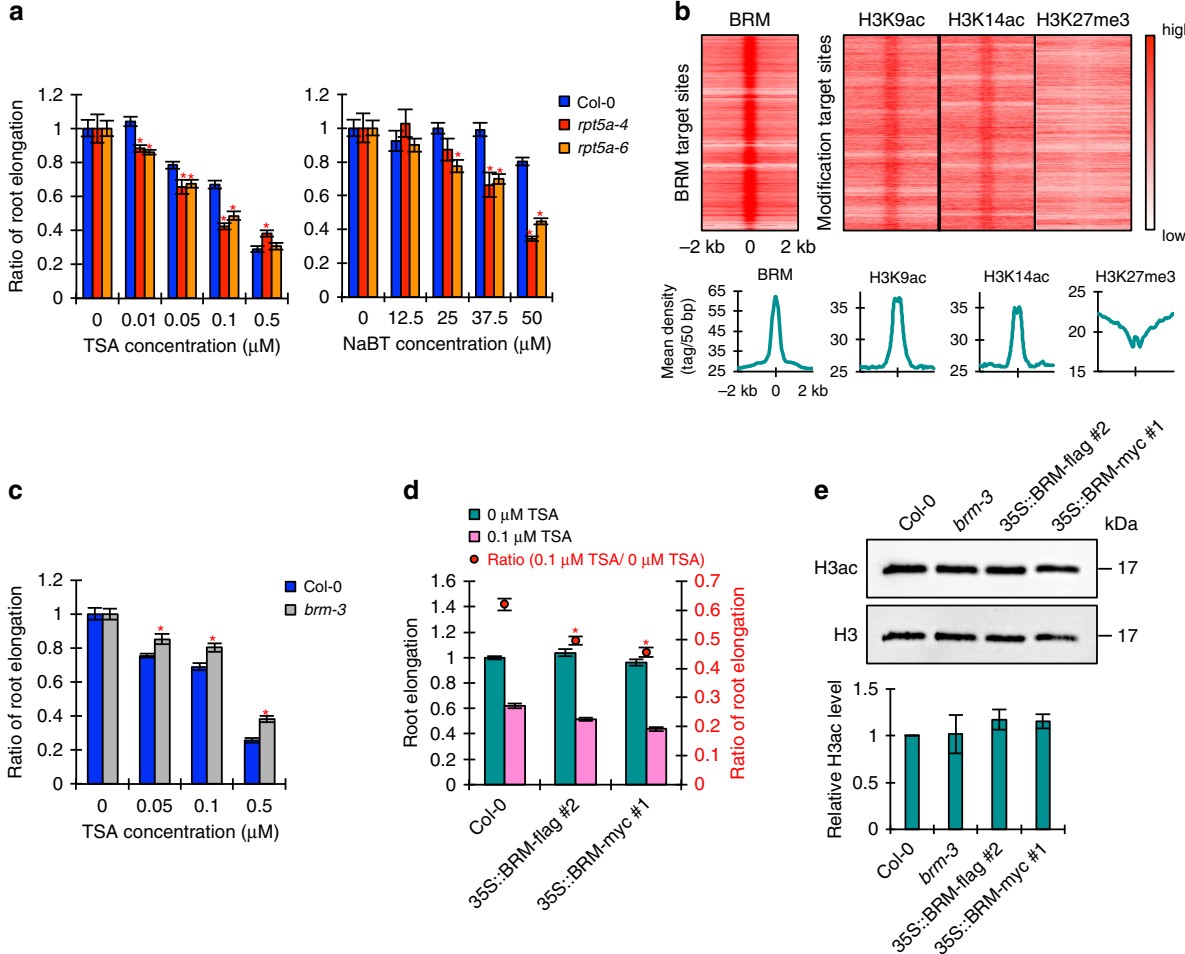

**Fig. 10** The negative effects of histone hyperacetylation are dependent on the BRM level. **a** Sensitivity of Col-0 and *rpt5a* mutant root growth to histone hyperacetylation caused by TSA or NaBT. Values are ratios relative to the values under each normal condition ($n = 15$–19, average ± s.e.m.; *$P < 0.05$, compared with Col-0, Student's *t*-test). **b** Heat map representation of the co-occupancy of BRM and histone modifications in the genome. Each horizontal line represents a BRM- or histone modification-enriched region, and the signal intensity shows the strength of BRM binding to those regions. Columns show the genomic region surrounding each target peak summit. Signal intensity is indicated by the shade of red. The lower panels show the mean density of BRM occupancy at sites of BRM binding and the indicated histone modifications. **c** Sensitivity of Col-0 and *brm-3* root growth to histone hyperacetylation caused by TSA. Values are ratios relative to the values under each normal condition ($n = 25$–27, average ± s.e.m.; *$P < 0.05$, compared with Col-0, Student's *t*-test). **d** Sensitivity of root growth of Col-0 and *BRM* high-expression plants to histone hyperacetylation caused by TSA. Values for root elongation are ratios relative to the value of Col-0 under 0 µM TSA conditions. Red circles represent ratios relative to the value under 0 µM TSA conditions ($n = 24$, average ± s.e.m.; *$P < 0.05$, compared with Col-0, Student's *t*-test). **e** Histone acetylation levels in roots of *brm-3* and *BRM* high-expression plants under normal B conditions. The lower panel shows the relative level of acetylated histone H3 accumulation normalized against total histone H3 ($n = 4$, average ± s.e.m.)

induced DSBs in roots result from attack by endogenous DNA-damaging factors of destabilized chromatin resulting from increased histone acetylation. It should be noted that excess ROS generation is often seen in high-B stressed plants[4]. The *rpt5a* mutants did not show enhanced sensitivity to ROS induction (Supplementary Fig. 1), but they did to further chromatin destabilization by treatment with HDAC inhibitors (Fig. 9b). These results indicate that chromatin destabilization via hyperacetylation is a principal action of B in the formation of DSBs.

Two copies of the genes for most plant RP subunits exist in the *Arabidopsis* genome, raising the possibility that plants assemble multiple 26SP forms composed of different subunits and that these different forms have unique properties and/or substrate specificities[38,39]. Unique functions of individual paralogs have been reported in *Arabidopsis*, which supports this hypothesis[17,18]. Our findings regarding the involvement of different subunits in

response to high-B stress indicate distinct roles for B-toxicity tolerance among paralogs and the possibility that 26SPs containing one or multiple specific RP subunits (at least among RPN2a, RPN8a, RPT2a, and RPT5a) are assembled to exert unique functions, especially in ameliorating high-B-induced DSBs in the RAM.

Our proteomic analysis of poly-Ub proteins and biochemical analyses identified BRM as a target of a 26SP of specific composition in response to high-B stress in roots (Supplementary Table 1 and Fig. 6). Moreover, genetic analysis demonstrated that BRM degradation is a key process for ameliorating DSBs as it prevents the enhancement of genomic instability caused by histone hyperacetylation and consequently helps maintain RAM organization (Figs. 7–10). These findings provoke a new question: how does the molecular action of BRM cause hypersensitivity to high-B stress? BRM has ATPase activity, which is required for the

remodeling activity of BRM-containing SWI/SNF chromatin remodeling complexes[24]. Given that ATP-dependent chromatin remodeling complexes promote susceptibility of nucleosomes to nucleases in vitro[40], an excess of BRM-containing SWI/SNF chromatin remodeling complexes might be a cause of extra DNA cleavage. Interestingly, brm-3 expresses a truncated BRM that lacks the C-terminal 454 amino acids containing the nucleosome interaction modules and bromodomain but that maintains chromatin remodeling activity[24]. However, brm-3 showed hypertolerance to high-B stress, suggesting a critical involvement of the BRM nucleosome interaction modules and bromodomain in the determination of sensitivity to high-B stress (Fig. 7b–e). It has been reported that the yeast bromodomain of SWI/SNF functions to anchor the complex on acetylated nucleosome arrays, leading to retention of the complex on chromatin[41]. Although it remains to be determined whether BRM directly binds to nucleosomes containing acetylated histone residues, our analysis of the genome-wide distributions of BRM and histone modifications indicated the enrichment of BRM around sites of histone H3K9ac and H3K14ac, suggesting the preferential binding of BRM to these acetylated histone residues in Arabidopsis (Fig. 10b). Taking our results together, we hypothesize that in 26SP mutants treated with high-B, histone hyperacetylation leads to chromatin opening in certain genomic regions, and that this loosened chromatin state can be maintained by the binding of non-degraded BRM through interaction of the bromodomain and acetylated histone residues, resulting in prolonged chromatin destabilization and the consequent formation of extra DSBs (Supplementary Fig. 14).

Notably, even under normal B conditions, the 26SP mutants displayed signs of increased DNA damage such as dead cells around the stem cell niche and higher levels of DSBs (Figs. 3 and 4). Based on the model described above, simultaneous high accumulation of histone acetylation and BRM causes genome instability even under normal B conditions, which would explain the observed DNA damage in the 26SP mutant. Consistent with this idea, we observed a reduction of a sign of DNA damage, dead stem cells, after impairing the function of BRM in the rpt5a mutant under normal B conditions (Fig. 7c). Therefore, we speculate that maintaining BRM function and histone acetylation at proper levels is crucial not only for the B-stress response but also for the normal development of plants.

It is possible that other candidate 26SP targets are involved in high-B stress tolerance (Supplementary Table 1). A phosphatidylinositol 4-kinase (PI4K) encoded by the AT1G49340 gene has been reported to be a potential 26SP target and to phosphorylate RPN10, a subunit of 19S RP[42]. The effect of RPN10 phosphorylation on 26SP activity is still unclear. However, in animal cells, phosphorylation of different 19S RP subunits, Rpt3 and Rpn6, likely causes differences in the substrate set of the 26SP[43]. These findings lead us to speculate that phosphoregulation of the 19S RP by PI4K affects its target recognition upon high-B stress. Interestingly, our candidate list included one of six homologues of the SIN3-like (SNL) protein, SNL5, a component of the SIN3 complex that includes class I HDACs, e.g., HAD6/HDA19. This complex acts as a co-repressor of transcription by reducing histone acetylation by HDAC, and its activity is upregulated by SUMOylation of subunits of the complex[44]. The biological significance of excess accumulation or poly-ubiquitination of SNL5 in the regulation of HDAC activity should be investigated in the future to evaluate its involvement in the increased histone acetylation levels in 26SP mutants.

Condensin II is another essential complex for the amelioration of high-B-induced DSBs[7]. It is, therefore, plausible that specific compositions of the 26SP and condensin II have concerted roles

in dealing with B genotoxicity. Our genetic analysis of a double mutant lacking the function of both complexes in the same pathway showed that high-B sensitivity was not additive in the double mutant compared with either single mutant (Supplementary Fig. 15). Interestingly, yeast condensin physically compacts rDNA arrays in response to nutrient starvation to maintain their stability[45,46]; however, it is unknown whether this condensin function is conserved in other eukaryotes including plants. Considering that histone deacetylation is crucial for the loading of the condensin complex onto chromatin in human cells[47], it is tempting to speculate that the histone hyperacetylation caused by high-B stress prevents or eliminates chromatin loading of the condensin complex, resulting in decompaction of chromatin and increased susceptibility to DNA-damaging factors. The 26SP might function to stabilize the condensin complex on chromatin by reducing histone hyperacetylation.

The 26SP and BRM, as well as condensin II, are widely conserved among eukaryotes. Furthermore, high-B-induced DSBs in association with high histone acetylation levels were also seen in human cells (Fig. 9). These findings lead us to speculate that DSB formation caused by a high intracellular B concentration occurs via common molecular mechanisms among eukaryotes.

## Methods

**Plant materials and growth conditions**. The heb3-1, heb6-1, and heb7-1 mutants of Arabidopsis (Col-0) were isolated as described previously[7]. The mutants were backcrossed to Col-0 three times. The T-DNA insertion mutants used in this study are listed in Supplementary Table 2. The T-DNA insertions in the homozygous mutants were confirmed by PCR using the primer sets listed in Supplementary Table 3. In all experiments, seeds were sown on media containing MGRL solution, 1% (w/v) sucrose and 1.5% (w/v) gellan gum[7]. Boric acid was used to adjust the B concentration in the medium. After 3 days incubation at 4 °C, the plates were placed vertically in a growth chamber (16-h light/8-h dark cycle, 22 °C) until analysis.

**Transplant method**. Five-day-old seedlings pre-incubated on vertically oriented normal MGRL plates were transferred to plates containing various concentrations of B, zeocin (Invitrogen, Carlsbad, CA, USA), MG132 (Peptide Institute, Ibaraki, Osaka, Japan), TSA (Wako, Osaka, Osaka, Japan), and NaBT (Wako) and their primary root-tip positions were marked. For γ-irradiation, the transferred 5-day-old seedlings were exposed to γ-rays derived from $^{137}$Cs at a dose rate of 0.83 Gy/min. After 4 or 7 days further incubation, the lengths of the newly elongated primary roots from the marked positions were determined using the ImageJ software (http://rsb.info.nih.gov/ij/).

**Gene expression analysis**. Total RNA from root tips (~1 cm from the tip) was extracted as described previously[7]. Quantitative real-time RT-PCR was performed with THUNDERBIRD® SYBR® qPCR Mix (Toyobo, Osaka, Osaka, Japan) on an ABI PRISM® 7300 (Thermo Fisher Scientific, Waltham, MA, USA). Actin8 was used for the normalization of cDNA concentration. The primers used are listed in Supplementary Table 3.

**Proteomic analysis of poly-Ub proteins**. A CycLex® Poly-Ubiquitinated Protein Enrichment & Detection Kit (MBL, Nagoya, Aichi, Japan) was used to purify poly-Ub proteins from roots. Total soluble proteins were extracted from whole roots (>4000 plants) in the provided extraction buffer with complete protease inhibitor cocktail (Roche, Basel, Switzerland). The extracts were centrifuged at 8000g for 15 min at 4 °C. The protein concentrations of the supernatants were determined using a Dc protein assay kit (Bio-Rad, Hercules, CA, USA). Twenty-five micrograms of extracted proteins (2 mg/ml) were incubated with 650 μl of GST-hHR23B-UBA domain-fused resin (provided in the CycLex kit) for 4 h at 4 °C. The proteins that bound to the resin were collected following the manufacturer's protocol and were extracted in 200 μl 2× SDS sample buffer [4% (w/v) SDS, 10 mM EDTA (pH 8.0), 40 mM Tris–HCl (pH 6.5), 10% (v/v) glycerol, 2% (v/v) β-mercaptoethanol, 1 mM PMSF]. The protein concentrations of the extracts were determined using Pierce® 660 nm protein assay reagent (Thermo Fisher Scientific).

Six micrograms of purified proteins were separated on a Perfect NT Gel M 7.5% (DRC, Tama, Tokyo, Japan) by SDS-PAGE, and subjected to in-gel trypsin digestion as described previously[48]. The obtained peptides were analyzed with LC-MS/MS (LTQ-Orbitrap XL, Thermo Fisher Scientific) and with the MASCOT server against a protein database from the National Center for Biotechnology

Information. The MASCOT search parameters were as follows: (i) taxonomy: *Arabidopsis*; (ii) potential modifications: carbamidomethyl (C) and GlyGly (K) as fixed modifications, and oxidation (M) as a variable modification; (iii) max missed cleavage: 1; (iv) peptide MS tolerance: ±10 ppm; and (v) fragment mass tolerance: ± 0.5 Da. See Supplementary Data 1 for lists of all detected proteins. The presence of putative motifs in the identified proteins was analyzed using the Eukaryotic Linear Motif resource (ELM; http://elm.eu.org/)[49] for D-box and KEN-box motifs, and the EMBOSS explorer (epestfind; http://emboss.bioinformatics.nl/cgi-bin/emboss/epestfind) for PEST sequences.

**Immunoblot analysis**. To detect poly-Ub proteins and RPT3 in roots, proteins were extracted using PBS with complete protease inhibitor cocktail and 1 mM PMSF. The extract was filtered through two layers of Miracloth (Millipore, Temecula, CA, USA). Protein concentration was determined with a Bio-Rad Protein kit (Bio-Rad). The extracts were mixed with equal amounts of 2× SDS sample buffer [4% (w/v) SDS, 10 mM EDTA (pH 8.0), 40 mM Tris–HCl (pH 6.5), 10% (v/v) glycerol, 2% β-mercaptoethanol, 1 mM PMSF, 0.02% bromophenol blue (BPB)] and boiled for 10 min. To detect histones in roots, proteins were extracted using 1× SDS sample buffer (without BPB). To detect BRM in roots, proteins were extracted using 1× SDS sample buffer (without BPB, with complete protease inhibitor cocktail and 10 μM MG132). The extract was filtered through two layers of Miracloth and then boiled for 10 min for histones or not boiled for BRM. The extract concentrations were relatively normalized by both the intensity of CBB-stained representative proteins separated by SDS-PAGE and by the amount of actin, as detected by immunoblotting.

For immunoblotting, 20 μg of extracted proteins (for the detection of poly-Ub proteins and RPT3) or relatively equal amount of extracted proteins (for the detection of histones and BRM) were separated by SDS-PAGE and transferred onto PVDF membranes. The antibodies used in this study were anti-poly-Ub proteins (FK1, Nippon Biotest Laboratories, Kokubunji, Tokyo, Japan) (1:1000 dilution), anti-RPT3[50] (1:5000 dilution), anti-α-tubulin (ab11304, Abcam, Cambridge, MA, USA) (1:5000 dilution), anti-histone H3 (MABI0301, MBL) (1:2000 dilution), anti-acetylated histone H3 (06-599, Millipore) (1:1000 dilution), anti-H3K9me1 (ab8896, Abcam) (1:1000 dilution), anti-histone H3K4me3 (ab6002, Abcam) (1:1000 dilution), and anti-BRM-N[51] (1:1000 dilution). The target proteins of each antibody were visualized with Western Lightning Reagent Plus-ECL (PerkinElmer, Norwalk, CT, USA) or ImmunoStar (Wako) on a C-DiGit Chemiluminescent Western Blot Scanner (MS Technosystems, Yodogawa, Osaka, Japan). To quantify the protein levels, the signal intensities of target bands were measured using Gel Analyzer in the Image J software. Uncut blot images are shown in Supplementary Figs. 16, 17 and 18.

**Cell culture and stimulation**. HeLa cells were maintained in low-glucose Dulbecco's Modified Eagle's Medium (DMEM) (Wako) supplemented with 10% (w/v) fetal bovine serum (FBS) at 37 °C under a humidified air/5% $CO_2$ atmosphere. For chemical treatment, cells were grown in 6-well plates to 20–25% confluency, and stimulated with B at 0, 0.5, 1.0, 2.5, or 3.75 mM, or with TSA at 10 nM for 48 h.

**Comet assay**. A CometAssay® Reagent Kit for Single Cell Gel Electrophoresis Assay (Trevigen, Gaithersburg, MD, USA) was used to perform a comet assay on plant cells as described previously[52]. For mammalian cultured cells, cells on 6-well plates were washed twice with PBS. After removing the PBS, 100 μl trypsin was added to the cells and incubated for 5 min at 37 °C. The trypsin reaction was stopped by adding 1 ml of 10% (v/v) FBS containing DMEM. Then, the cell mixture was transferred to a 1.5-ml tube. The cells were harvested by centrifugation at 800$g$ for 1 min and resuspended in 1 ml PBS. Slides were prepared following the manufacturer's protocol. Images of nuclei were obtained by inverted fluorescence microscopy on a BX53 microscope (Olympus, Shinjuku-ku, Tokyo, Japan) equipped with a DOC-CAM HR CCD camera (Molecular Devices, Chuo-ku, Tokyo, Japan). Images were analyzed using the ImageJ software plugin CometAssay distributed by the University of North Carolina School of Medicine, USA (https://www.med.unc.edu/microscopy/resources/imagej-plugins-and-macros/comet-assay).

**Confocal imaging**. To visualize the root tip structures of the wild type, mutants, and GFP-expressing transgenic plants, plants were stained with PI (10 μg/ml; Molecular Probes, Eugene, OR, USA) for 5 min, and images were captured using a confocal laser scanning microscope (FV-1000; Olympus) with excitation wavelengths of 488 nm for GFP and 559 nm for PI. The counting of meristematic cortex cells was performed using the ImageJ plugin Cell Counter (https://imagej.nih.gov/ij/plugins/cell-counter.html).

**Proteasome activity**. To extract the 26SP fraction, whole roots (~0.1 g) were homogenized in liquid nitrogen by grinding to a fine powder using a mortar and pestle. Then, the homogenized material was resuspended in 100 μl extraction buffer (50 mM Tris–HCl pH 7.5, 20% (v/v) glycerol, 1 mM EDTA pH 8.0, 10 mM β-mercaptoethanol) and centrifuged at 15,000$g$ for 15 min at 4 °C. The protein concentrations of the supernatants were determined with a Bio-Rad Protein kit (Bio-Rad). Caspase-like and chymotrypsin-like proteasome activities were

measured with fluorogenic peptides, Z-LLE-AMC and Suc-LLVY-AMC, respectively (Peptide Institute, Minoh, Osaka, Japan). The extract (5 μl) was mixed with reaction buffer [100 mM Tris–HCl pH 8.0, 0.02% (w/v) SDS] containing 50 μM fluorogenic peptides (95 μl) in DMSO or MG132 (Wako), an inhibitor of caspase-like and chymotrypsin-like activities. The mixture was incubated at 37 °C. After pre-incubation for 5 min, the fluorescence of AMC released from the digested peptides was measured on a fluorometer (Infinite® M1000, Tecan Trading AG, Mannedorf, Switzerland) every 1 min for 20 min (excitation of 380 nm and emission of 460 nm). The absolute amount of released AMC was calculated based on a standard curve generated with a series of dilutions of free AMC (Peptide Institute). Then, the rate of increase of AMC release in each sample was calculated. The value obtained using MG132-containing buffer was subtracted from the value obtained using DMSO-containing buffer. Finally, the values were normalized against each protein concentration and presented as proteasome activity.

**Genome-wide distribution of BRM and histone modifications**. All sequenced reads of BRM were obtained as a fastq file from the Sequence Read Archive under accession SRX1184288[32], and mapped on the *Arabidopsis* genome reference (TAIR10) using the bowtie 1.2.2[53] (Langmead) software with the following parameters: '-m 1 -p 4'. The peak summits were obtained as bed files from the Gene Expression Omnibus under accession GSE72736 for BRM and H3K27me3[32] and GSE89768 for H3K9ac and H3K14ac[33]. Heat maps and values for the distribution of mean density were obtained using the seqMINER software[54] with default parameters and K-means enrichment linear normalization methods.

**Positional mapping of *HEB* genes**. *rpn8a-3/heb3-1*, *rpt5a-5/heb6-1*, and *rpt5a-6/heb7-1* mutants (M3 generation) (Col-0 background) were crossed with Ler wild-type plants. DNA was extracted from shoots of 2925 (for *rpn8a-3/heb3-1*), 25 (for *rpt5a-5/heb6-1*), and 2243 (for *rpt5a-6/heb7-1*) F2 plants and was genetically analyzed as described previously[7]. The simple sequence length polymorphism and cleaved-amplified polymorphic sequence markers are listed in Supplementary Table 4. The short-root phenotype of F3 progeny of F2 recombinants was used to map the *heb* mutations.

**Generation of transgenic plants**. For complementation tests of the *heb* mutants, genomic fragments of *RPN8a* and *RPT5a* were amplified from Col-0 genomic DNA and subcloned into pENTR-D/TOPO following the manufacturer's protocol (Invitrogen). Amplified fragments contained 2735 and 1061 bp upstream of the start codon of *RPN8a* and *RPT5a*, respectively, as promoter regions. The cloned genomic fragments were subsequently subcloned into a Gateway destination vector, pMDC107, containing a *GFP* gene[55] by LR recombination with LR clonase II following the manufacturer's protocol (Invitrogen). For enhanced-expression of *BRM*, *BRM* coding sequence was amplified from a cDNA derived from Col-0 roots. The *BRM* fragment was inserted into pENTR-D/TOPO and then subcloned into pGWB511 or pGWB517 containing flag or myc tags[56], respectively as described above. Primers used are listed in Supplementary Table 3. The constructs were mobilized into *Agrobacterium tumefaciens* (GV3101 pMP90) and used to transform plants by the floral dip method. Transgenic plants were selected on medium containing 1/2× Murashige and Skoog, 1% sucrose, 20 μg/ml hygromycin B, and 250 μg/ml claforan. T3 transgenic plants harboring homozygous T-DNA were used for subsequent analyses.

**MNase assay**. For the isolation of nuclei, whole roots (~0.5 g) were homogenized in liquid nitrogen by grinding to a fine powder using a mortar and pestle. Then, the homogenized material was resuspended in 5 ml of nuclear isolation buffer (NIB) (10 mM Tris–HCl pH 7.5, 1.14 M sucrose, 5 mM MgCl₂, 5 mM β-mercaptoethanol). The suspension was filtered through two layers of Miracloth (Millipore) and centrifuged at 2500$g$ for 5 min at 4 °C. The pellet was resuspended in 5 ml of cold NIB containing 0.15% Triton X-100, incubated on ice for 15 min, and centrifuged at 2500$g$ for 5 min at 4 °C. The pellet was resuspended in 500 μl of micrococcal nuclease (MNase) digestion buffer (50 mM Tris–HCl pH 8.0, 0.3 M sucrose, 5 mM MgCl₂, 1.5 mM NaCl, 0.1 mM CaCl₂, 5 mM β-mercaptoethanol). Digestion was performed with 2.5 U/ml MNase (TAKARA, Kusatsu, Shiga, Japan). The MNase (5 U/ml)-containing MNase digestion buffer (400 μl) was added to the nuclei suspension (400 μl), and the mixture was incubated at 37 °C (final concentration, 2.5 U MNase). Aliquots (50 μl) were removed from the reaction mixture at the indicated time points and the reaction was stopped by adding 2× stop buffer (50 mM EDTA pH 8.0, 1% SDS, 0.2 mg/L Proteinase K). The samples were incubated for 3 h at 37 °C to degrade protein. The results were visualized by electrophoresis on a 1% agarose gel stained with ethidium bromide. The images were captured using an E-BOX VX2 (MS Instruments, Yodogawa, Osaka, Japan) and the intensity of non-digested chromatin was analyzed using ImageJ.

## Data availability

The authors declare that all other data supporting the findings of this study are available within the Article and its Supplementary Information files or are available from the corresponding authors upon request.

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

## Acknowledgements

We thank Yuko Kawara (The University of Tokyo), Sayoko Mibu (Tokyo University of Science), and Yuka Asako (Tokyo University of Science) for technical assistance, Masayuki Fujiwara for technical advice (Keio University), and José C. Reyes (Andalusian Molecular Biology and Regenerative Medicine Centre) for the BRM antibody and *brm-3* mutant. This work was supported by CREST and JST grants to S.M. and JSPS KAKEHNHI grants to S.M. [Nos. JP15H05962 and JP15H05955], T.S. [No. JP25113002], and T.F. [No. JP25221202].

## Author contributions

T.S.: conception and design, acquisition of data, analysis and interpretation of data, drafting of the article. N.S., S.M., T.F.: conception and design, interpretation of data. Y. T.-I., T.H., T.M.M., Y.F.: acquisition of data.

## Additional information

**Competing interests:** The authors declare no competing interests.

