## [Peer Review File · Nature Communications]

Reviewers' comments:

Reviewer #1 (Remarks to the Author):

The authors identified four regulatory particle (RP) subunits of 26S proteasome (26SP) that are required for the tolerance to high boron stress. Phenotypic analysis of the mutants suggested that the identified RP subunits function in maintaining chromatin stability by preventing DSBs rather than in repair of damaged DNA. Furthermore, the chromatin remodeling factor BRAHMA (BRM) was identified as a target of the 26SP by proteome analyses of poly-ubiquitinated proteins and BRM level was correlated with the sensitivity of plants to high boron. In addition, high boron stress enhanced acetylation of histones. It was suggested that the 26SP prevents the excessive function of BRM in chromatin remodeling accompanied by histone hyperacetylation, resulting in maintenance of chromatin stability to avoid the formation of DSBs under high boron conditions.

Although the manuscript provides interesting data about the involvement of 26SP in the tolerance to high boron stress, the molecular mechanism is not defined. Furthermore, the role of BRM and its relationship with histone hyperacetylation in the tolerance to high boron stress are not clear documented, additional information is needed to support and/or clarify the authors' conclusions.

Specific comments:

1. Page 10, line 194 - High boron stress significantly reduced the number of meristematic cortical cells among the wild type and all high boron-hypersensitive RP subunit mutants with much reduced levels in the mutants (less than 28% inhibition in the wild type and more than 36% inhibition in the mutants) (Figure 3B and Supplemental Figure 4G). These results indicated that the RP subunits are crucial for the maintenance of RAM organization especially under high boron conditions.

It is important to analyze the genes and pathways that are affected in the mutants responsible for the phenotypes.

2. Page 14, line 313 - Four out of eight transgenic lines exhibited poorer root growth under high boron conditions but not under normal conditions than the wild type (Figure 7A), suggesting that BRM accumulation has an inhibitory effect on root elongation, especially under high boron conditions.

It is hard to make any conclusions if only 50% transgenic lines display the root growth phenotype. The authors should examine the expression levels of BRM in transgenic lines and establish whether there is a correlation between the BRM level and phenotype in the transgenic lines. I suggest that the authors should also analyze the phenotypes of the BRM overexpressing lines in Figure 7c-7e and Figure 8.

3. Figure 7. Regulation of BRM levels through the 26SP containing RPT5a is crucial for RAM maintenance under high boron conditions.

Again, it is important to analyze the genes and pathways that are affected in the mutants responsible for the phenotypes.

4. Page 16, line 347 - The inhibition of root growth (Figure 8B) and the reduction in RAM size (Figure 8C and 8D) by zeocin treatment were less pronounced in *brm-3* than those in the wild type. Moreover, cell death in the stem cell niche caused by zeocin inducible-DSBs was suppressed in *brm-3* (Figure 8C). These results indicated that BRM has negative effects on the suppression of DSBs caused by various DNA damaging factors, including high boron stress.

The molecular mechanisms of why BRM has negative effects on the suppression of DSBs need to be analyzed.

5. Page 17, line 376 - Simultaneous treatment of boron and TSA resulted in higher ratio of root elongation in both the wild type and *rpt5a-4* compared with single treatment of boron (Figure 9C), indicating that boron and TSA act on the common target in Arabidopsis roots. Overall, these data demonstrated that high boron stress induces histone hyperacetylation, which is likely to be a cause of high boron-hypersensitivity in the *rpt5a* mutants.

6. More direct evidences are required to support the conclusion that high boron stress induces histone hyperacetylation is likely to be a cause of high boron-hypersensitivity in the *rpt5a* mutants. I suggest that the authors can analyze histone deacetylase mutants to test if these mutants with histone hyperacetylation also have the boron-hypersensitivity phenotype.

7. Only one *brm* mutant allele (*brm-3*) was used for phenotype analysis. I suggest that the authors can use additional mutant allele.

Reviewer #2 (Remarks to the Author):

The manuscript described Arabidopsis mutants with hypersensitivity to excess boron and showed that degradation of a SWI/SNF subunit BRM by 26S proteasome is important for ameliorating high-B inducible DSB. The authors also observed that high-B treatment induces histone hyperacetylation and a positive correlation between the histone acetylation and DSB formation. The manuscript contains interesting information since it suggests that high-B inducible DSB is integrated at the level of the histone acetylation. However, some conclusions are not justified.

1, The authors suggest that the growth defects caused by *heb* mutations are specific to high-B stress. However, the data in Figure 1c showed that *heb6* and *heb7* mutants seem hypersensitive to -B and *heb* mutants are insensitive to As, suggesting that growth phenotypes of *heb* mutants are not specific to high-B stress.

2, In Figure 2f, it seems that the root elongation of RPT5a-GFP;*heb6* transgenic lines grown on medium with 0.03mM B was similar to *heb6*. Could RPT5a-GFP complement the short root phenotype of *heb6* or *heb7*? I would like to suggest the authors to show the images of RPT5a-GFP;*heb6* and RPT5a-GFP;*heb7* transgenic lines. If RPT5a-GFP could not complement the short root phenotype of *heb6* or *heb7* seedlings grown on medium with 0.03mM B, *heb6* or *heb7* might contain other mutations.

3, BRM protein accumulation was higher in *rpt5a* mutants than that in the wild type (Figure 6c). BRM accumulation was further promoted by high-B stress only in the *rpt5a* mutants (Figure 6c). If accumulation of BRM is a main cause of the *rpt5a* hypersensitivity to high-B stress, *brm* loss of function mutant could suppress the the phenotype of the *rpt5a* in response to high-B stress. Surprisingly, the root elongation ratio of *rpt5a brm* double mutant was similar to that of *rpt5a* under high-B stress (Figure 7b), suggesting that accumulation of BRM is not a main reason of the *rpt5a* hypersensitivity to high-B stress.

In Figure 7d, genetic interaction between *rpt5a* and *brm* seems additive for the number of meristematic cortex cells under high-B conditions, suggesting that RPT5a and BRM might act redundantly.

4, It has been reported that the yeast bromodomain of SWI/SNF functions to anchor the complex on acetylated nucleosome arrays, leading to retention of the complex on chromatin. Does Arabidopsis BRM bind to acetylated histone? Does BRM affect the level of histone acetylation?

5, Multiple histone modifications could affect chromatin structure. Does high-B stress specifically affect levels of histone acetylation? Does high-B stress influence levels of histone methylation or other modifications?

6, In Figure 5a, it is hard to observe differences of poly-Ub proteins between the wild type and mutants. In this case, it would be more reliable to do replicates and show statistics.

Reviewer #3 (Remarks to the Author):

In this work, the authors follow up on previous observations that high B toxicity is manifested through double strand DNA breaks (Plant Cell 25:3533), and provide a substantial advance in the mechanism through which this occurs. Specifically:

- They have genetically mapped Arabidopsis boron hypersensitivity mutants (heb). They show convincingly that the defects are the result of disruption of specific regulatory subunits of the 19S regulatory particle of the 26S proteasome.
- They also show that only certain paralogs of these proteins affect B sensitivity that argues for subfunctionalization of the regulatory particle subunits.
- By using MS analyses of poly Ub proteins isolated from mutant and wild type plants under high B stress, they have identified 13 proteins that are preferentially represented in the rpt5a-4 mutant background, and which are presumably targets for proteasome turnover during high B.
- They have focused on one of these, BRM which encodes a chromatin remodeling ATPase. It is shown that BRM over accumulates in rpt5a-4 plants, and is at least in part responsible for the double stranded breaks and enhanced B toxicity phenotype of these plants.
- Finally, they provide a link between histone hyperacetylation driven by boron (presumably from the inhibition of histone deacetylases), BRM changes in chromatin conformation, which is postulated to lead to increased DNA damage.

This is a substantial body of work, carefully executed, and provides new insights into the molecular basis of boron toxicity through DNA damage. Further, while the work is principally carried out on plants, the authors make a case that this a general mechanism for boron toxicity across species, including humans. However, the authors need to address the following technical and conceptual points:

1. While the authors make a compelling case for a link between BRM accumulation and B toxicity/double stranded breaks, this does not seem to be the complete story. They need to explain the following:

- Wild type plants show B toxicity but show no apparent changes in the level of BRM3 in response to elevated B (Fig. 6C).
- The authors acknowledge on lines 335-338 that BRM is not the solo cause of hypersensitivity of

proteasome mutants to B. (Thus, they should correct the statement in the abstract on lines 36-37 that they "show the molecular basis of high B double strand breaks"). The MS data show that several poly Ub proteins besides BRM are preferentially accumulated in rpt5a-4 mutants. Some of these could also contribute to B toxicity. Have they been evaluated? At the very least they should be discussed in this context.

2. The relationship between high B, histone deacetylase activity, and double stranded breaks is a little bit difficult to follow in the author's narrative. Is the contention that the sole effect of high boron is the inhibition of histone deacetylase activity leading to enhanced chromatin binding? I.e., in this case borate or boric acid has no direct effect on introducing nucleic acid defects? The authors cover this in the discussion but a model would be helpful here. What is the source of DNA damaging/double stranded breaks? Can other effects besides this potential inhibition of Histone deacetylases be at play?

3. In their model, the authors should put condensin II into context. In the discussion on pages 21-22, lines 479-484 and SFig. 11, they stated that the double mutant of (cap-g2-2 and rpt5a-4) demonstrate a concerted role in dealing with B genotoxicity. I do not see much difference between single and double mutants in SFig. 11 and the relationship between condensin and 26SP is not clear.

4. On lines 376-379, the authors state that "Simultaneous treatment of B and TSA resulted in high ratio of root elongation in wild type and rpt5a-4 compared with single treatment of B" and take this to indicate that B and TSA have a common target. I do not see how this conclusion follows from the data, and the two agents can have distinct or overlapping targets.

5. Heb6-1 and 7-1 show substantial inhibition of growth, even under normal B conditions. Are the enhanced B toxicity effects simply due to the fact that these plants are already compromised physiologically?

6. Lines 254-265 and Figure 5:

- The authors state that there is a higher accumulation of poly-Ub proteins in rpt5a but not rpt5b mutants. This is not readily apparent from Fig. 5a and there is no quantitation presented.
- Are the proteolytic activities shown in Fig. 5b solely due to 26S protease activities? It seems as if this could be a general proteolytic activity. Can we conclude that these activities are Ub dependent? (as stated in lines 261-2).
- The authors state that the 26S proteolytic activity is a requirement for high B tolerance. From Fig. 5b it is not clear that 26S proteolytic activity is elevated by high B conditions.

7. It would be appropriate to include more information in the supplementary data regarding the MS data including the numbers of peptides detected, sequence ranges, and the proteins identified in each replicate.

8. For clarity, the authors should indicate the stem cell niche regions in the PI stained micrographs (e.g., Fig. 3a, 7c, 8c). A high magnification micrograph would be informative in terms of showing this region and the effects on meristem cell organization.

9. For Western blot data (e.g., Fig. 6b and c, and Fig. 9 a and h), reproducibility and error bars on histograms are needed. This is especially important since some of the differences (e.g., Fig. 6b and Fig. 9) are not large (less than 2-fold).

10. On page 17, lines 384 and 385, it is stated that "Histone hyperacetylation is a well-known cause of interphase chromatin decondensation, resulting in impaired chromatin vulnerability...". Do they mean "enhanced" instead of "impaired"?

11. Although not a major point of the study, SFig 6 shows that *rpn8a* and *rpn5a* mutants accumulate less B than wild type under elevated B conditions. Are transporters affected?

12. Is there a difference in the accumulation of ROS in mutant and wild type plants under normal or elevated B?

13. Minor comments:

- Fig. 6d is mislabeled.
- Fig. 8c has a random "B" that is not defined.
- For sodium butyrate (Fig. 9 and line 507) a different abbreviation besides "NaB" should be used since at first glance this appears to be "boron".
- Line 394 "were" instead of "was".
- Where appropriate include v/v or w/v in methods and materials and spell out all abbreviations (e.g., FPS and DMEM on line 591, there are others).

Reviewer #4 (Remarks to the Author):

The metalloloid element boron (B) is an essential nutrient for plant growth and development. However, in excess it is toxic, generating DNA damage and subsequently resulting in growth inhibition. Here, through a mutagenesis screen for plants being hypersensitive to B, two 26S proteasome (26SP) mutants were identified. Microscopic analysis demonstrated that both genes are essential to maintain root meristem function under high B concentrations. By performing a differential proteomics screen, the authors identified the BRAMA (BRM) chromatin remodeling complex protein as an 26SP target, and subsequently demonstrated that BRM overproducing and knockout plants display B resistance and sensitivity, respectively. As BRM is known to bind acetylated histone tails, the authors consequently analyzed histone acetylation levels in response to B treatment. Through a set of dedicated experiments, the authors demonstrated that B excess results in an accumulation of histone acetylation (likely through an inhibition of histone deacetylases), resulting in a more open chromatin structure that makes the DNA more accessible to DNA breaking events. It is hypothesized that BRM helps in opening of the chromatin through its chromatin remodeling activity.

This work represents a big step forward in the field. Although high B levels were demonstrated before to trigger a DNA damage response, here the authors convincingly identified the underlying reasons of this toxicity through the generation of a very complete dataset using a number of well-thought-through and clever experiments (e.g. the experiment that helps to discriminate a role for the 26SP in preventing DNA damage). In the end, the data represent an almost completely proven pathway for B toxicity. This excludes direct proof that B inhibits the activity of histone deacetylases, but then again there is sufficient indirect evidence for this supporting this part of the model.

Specific comments:

I would help to include a graphical model summarizing all data.

Figure 5a: It would help to include a quantification of the amount of Ub proteins.

Lines 305-308: The authors cannot exclude a difference in translation efficiency in low versus high B, thus delete statement.

Already in the absence of high B the 26SP mutant display signs of DNA damage (such as the dead stem cells and upregulation of DNA repair genes). The authors should speculate on the origin of this DNA damage. Is it also linked to more open chromatin structure? Likewise, I would like to see some speculation on by BRM is more targeted for destruction in high versus low B levels.

Figure 6: Panels are mislabeled.

Point-by-point Responses to the Reviewers' Comments

We appreciate the valuable comments from the four reviewers. We have carefully revised the manuscript in accordance with their comments and suggestions. The revised parts are in red in the revised manuscript, and detailed point-by-point responses are provided below.

Reviewer 1

The authors identified four regulatory particle (RP) subunits of 26S proteasome (26SP) that are required for the tolerance to high B stress. Phenotypic analysis of the mutants suggested that the identified RP subunits function in maintaining chromatin stability by preventing DSBs rather than in repair of damaged DNA. Furthermore, the chromatin remodeling factor BRAHMA (BRM) was identified as a target of the 26SP by proteome analyses of poly-ubiquitinated proteins and BRM level was correlated with the sensitivity of plants to high B. In addition, high B stress enhanced acetylation of histones. It was suggested that the 26SP prevents the excessive function of BRM in chromatin remodeling accompanied by histone hyperacetylation, resulting in maintenance of chromatin stability to avoid the formation of DSBs under high B conditions.

Although the manuscript provides interesting data about the involvement of 26SP in the tolerance to high B stress, the molecular mechanism is not defined. Furthermore, the role of BRM and its relationship with histone hyperacetylation in the tolerance to high B stress are not clearly documented, additional information is needed to support and/or clarify the authors' conclusions.

Response: Thank you for your constructive and valuable suggestions. Our revised version with additional experiments shows a more distinct relationship between histone hyperacetylation and high-B stress.

Specific comments:

Comment 1: Page 10, line 194 - High B stress significantly reduced the number of meristematic cortical cells among the wild type and all high B-hypersensitive

RP subunit mutants with much reduced levels in the mutants (less than 28% inhibition in the wild type and more than 36% inhibition in the mutants) (Figure 3B and Supplemental Figure 4G). These results indicated that the RP subunits are crucial for the maintenance of RAM organization especially under high B conditions.

It is important to analyze the genes and pathways that are affected in the mutants responsible for the phenotypes.

Response: We agree with your comment. Thus, we have shown that a defect in BRM function in the *rpt5a* mutant recovered the RAM morphology under high-B stress conditions as shown in Fig. 7c–e. We believe that these data are sufficient to demonstrate that BRM is one of the genes responsible for the phenotypes.

Comment 2: Page 14, line 313 - Four out of eight transgenic lines exhibited poorer root growth under high B conditions but not under normal conditions than the wild type (Figure 7A), suggesting that BRM accumulation has an inhibitory effect on root elongation, especially under high B conditions.

It is hard to make any conclusions if only 50% transgenic lines display the root growth phenotype. The authors should exam the expression levels of BRM in transgenic lines and establish whether there is a correlation between the BRM level and phenotype in the transgenic lines.

I suggest that the authors should also analyze the phenotypes of the BRM overexpressing lines in Figure 7c-7e and Figure 8.

Response: In accordance with the reviewer's suggestion, we examined the *BRM* mRNA levels in the transgenic lines and found a slight correlation between the *BRM* mRNA level and phenotype. We have added these data as Supplementary Fig. 10a. In this experiment, the *BRM* mRNA level was found to be only 2.5-fold higher in the transgenic lines. Therefore, we changed the term

“overexpression” to “enhanced expression” or “high-expression”.

We also analyzed the phenotypes of two representative transgenic lines with increased BRM expression in regards to RAM morphology, DSB accumulation, and sensitivity to another DSB-inducing chemical, zeocin. We have added these data as Supplementary Fig. 10b–e. Both lines showed a high reduction rate in RAM size and high accumulation of DSBs upon high-B stress. However, the increased BRM expression did not affect the sensitivity to zeocin treatment. One possible explanation for this phenomenon is a difference in the molecular action leading to DSB accumulation between high-B and zeocin. We have added an explanation of these results at P14 L334 – L335, and P16, L382 – P17, L387.

Comment 3: Figure 7. Regulation of BRM levels through the 26SP containing RPT5a is crucial for RAM maintenance under high B conditions.

Again, it is important to analyze the genes and pathways that are affected in the mutants responsible for the phenotypes.

Response: Thanks for your valuable suggestion. Our data have provided sufficient evidence to claim that *BRM* is responsible for the phenotypes. We understand that analyzing the genes affected by BRM over-accumulation in the *rpt5a* mutant under high-B stress conditions will allow a more detailed hypothesis. Such work would make it possible to identify the genes responsible for the maintenance of RAM morphology. However, we consider this work to be beyond the scope of our manuscript. We would like to mention that BRM is a determinant of the hypersensitivity of *rpt5a* to high-B stress in terms of both root elongation and morphology. As a next step, we plan to design a new project based on your constructive suggestion.

Comment 4: Page 16, line 347 - The inhibition of root growth (Figure 8B) and the reduction in RAM size (Figure 8C and 8D) by zeocin treatment were less

pronounced in *brm-3* than those in the wild type. Moreover, cell death in the stem cell niche caused by zeocin inducible-DSBs was suppressed in *brm-3* (Figure 8C). These results indicated that BRM has negative effects on the suppression of DSBs caused by various DNA damaging factors, including high B stress.

The molecular mechanisms of why BRM has negative effects on the suppression of DSBs need to be analyzed.

Response: Thank you for valuable comments. We have discussed the mechanism including its relationships with histone acetylation in the discussion section (P22, L520 – P23, L551).

Comment 5,6: Page 17, line 376 - Simultaneous treatment of B and TSA resulted in higher ratio of root elongation in both the wild type and *rpt5a-4* compared with single treatment of B (Figure 9C), indicating that B and TSA act on the common target in *Arabidopsis* roots. Overall, these data demonstrated that high B stress induces histone hyperacetylation, which is likely to be a cause of high B-hypersensitivity in the *rpt5a* mutants.

More direct evidences are required to support the conclusion that high B stress induces histone hyperacetylation is likely to be a cause of high B-hypersensitivity in the *rpt5a* mutants. I suggest that the authors can analyze histone deacetylase mutants to test if these mutants with histone hyperacetylation also have the B-hypersensitivity phenotype.

Response: Thanks to your suggestion, we noticed that there was a contradiction in concluding that histone hyperacetylation was likely a cause of high B-hypersensitivity in the *rpt5a* mutants. If this was true, TSA treatment should increase high-B sensitivity even in the wild type. However, the result was completely opposite as shown in Fig. 9b. Now, we consider the high-B hypersensitivity of *rpt5a* to be attributable mainly to the overfunction of BRM. In our new experiments, *brm-3* showed high tolerance to TSA treatment as shown

in Fig.10c. In addition, transgenic lines with enhanced BRM expression showed normal growth but were highly sensitive to TSA treatment (Fig. 10d). These facts indicated that the negative effects of histone hyperacetylation were dependent on BRM levels. Therefore, the increased BRM levels in the *rpt5a* mutants would act to exacerbate the negative effects of histone hyperacetylation on genome stability. Accordingly, we have revised our hypothesis, the order of the results and the conclusions in the section “High-B stress induces histone hyperacetylation” (P17, L389 – P18, L420). Moreover, we have created a new section “BRM enhances the negative effects of histone hyperacetylation“ at P19, L455 – P20, L479.

To further support our hypothesis that B and TSA act on a common target in *Arabidopsis* roots, we analyzed two histone deacetylase mutants that show increased histone acetylation genome-wide. These *hdac* mutants were found to be less sensitive to high-B stress (Fig. 9c), which is consistent with the results of simultaneous treatment of B and TSA. We have described this result at P17, L399 – L408.

Comment 7: Only one *brm* mutant allele (*brm-3*) was used for phenotype analysis. I suggest that the authors can use additional mutant allele.

Response: In accordance with your comment, we have analyzed *brm-1* and *brm-6* in addition to *brm-3*; however, these mutant alleles show severely inhibited root growth even under normal B conditions as previously reported by another group (Archacki et al., 2013, *PLoS one*). Therefore, it was impossible to evaluate the additional inhibitory effect of high-B stress on root growth using these alleles. We consider the high tolerance to high-B stress in *brm-3* and the high sensitivity to high-B stress in transgenic plants with enhanced expression of BRM to be sufficient evidence to conclude that the function of BRM is closely associated with the sensitivity to high-B stress.

Reviewer #2 (Remarks to the Author):

The manuscript described Arabidopsis mutants with hypersensitivity to excess B and showed that degradation of a SWI/SNF subunit BRM by 26Sproteasome is important for ameliorating high-B inducible DSB. The authors also observed that high-B treatment induces histone hyperacetylation and a positive correlation between the histone acetylation and DSB formation. The manuscript contains interesting information since it suggests that high-B inducible DSB is integrated at the level of the histone acetylation. However, some conclusions are not justified.

Response: Thank you for your positive comment that our manuscript includes interesting information for biology fields. We have strengthened the evidence for some of our conclusions in accordance with your comments.

Specific comments:

Comment 1: The authors suggest that the growth defects caused by *heb* mutations are specific to high-B stress. However, the data in Figure1c showed that *heb6* and *heb7* mutants seem hypersensitive to -B and *heb* mutants are insensitive to As, suggesting that growth phenotypes of *heb* mutants are not specific to high-B stress.

Response: In accordance with your suggestion, we have revised our arguments for the results and the specificity of the growth phenotypes of the *heb* mutants at P6, L134 – L136 and P6, L141 – L142.

Comment 2: In Figure 2f, it seems that the root elongation of RPT5a-GFP;*heb6* transgenic lines grown on medium with 0.03mM B was similar to *heb6*. Could RPT5a-GFP complement the short root phenotype of *heb6* or *heb7*? I would like to suggest the authors to show the images of RPT5a-GFP;*heb6* and RPT5a-GFP;*heb7* transgenic lines. If RPT5a-GFP could not complement the short root phenotype of *heb6* or *heb7* seedlings grown on medium with 0.03mM B, *heb6* or *heb7* might contain other mutations.

Response: As you have pointed out, we cannot exclude the possibility that other mutations caused retarded root growth under normal conditions in the *heb6* and *heb7* mutants, although they were backcrossed three times. In terms of relative root growth, we were able to complement the hypersensitivity to high-B stress in the *heb6* and *heb7* mutants by introducing RPT5a-GFP (Fig. 2e, f). In addition, we have added the pictures of root morphology to show that RPT5a-GFP also rescues the RAM morphology of *heb* mutants irrespective of B conditions (Supplementary Fig. 7). These additional data reinforce our argument that *RPT5a* is responsible for both the *heb6* and *heb7* mutants. We have added an explanation of the root morphology results at P10, L221 – L226.

Comment 3: BRM protein accumulation was higher in *rpt5a* mutants than that in the wild type (Figure 6c). BRM accumulation was further promoted by high-B stress only in the *rpt5a* mutants (Figure 6c). If accumulation of BRM is a main cause of the *rpt5a* hypersensitivity to high-B stress, *brm* loss of function mutant could suppress the phenotype of the *rpt5a* in response to high-B stress. Surprisingly, the root elongation ratio of *rpt5a brm* double mutant was similar to that of *rpt5a* under high-B stress (Figure 7b), suggesting that accumulation of BRM is not a main reason of the *rpt5a* hypersensitivity to high-B stress.

In Figure 7d, genetic interaction between *rpt5a* and *brm* seems additive for the number of meristematic cortex cells under high-B conditions, suggesting that RPT5a and BRM might act redundantly.

Response: As you have pointed out, the root elongation ratios under 3 mM B conditions indicate that BRM is not the sole cause of the *rpt5a* hypersensitivity to high-B stress (Fig. 7b). We have also shown that a defect in *brm* function suppresses the phenotypes of the *rpt5a* mutants in response to moderate high-B stress (1.5 mM B) (Fig. 7c–e). Moreover, we found that the severe disorder in meristematic cortex number and morphology of the RAM in *rpt5a* in response to 3 mM B was rescued by defective BRM function (Fig. 7d, e). These results

strongly support that BRM overfunction is “one of the main causes” of the *rpt5a* hypersensitivity to high-B stress. We have revised the sentence as follows “To examine whether BRM overfunction was one of the main causes of the *rpt5a* hypersensitivity to high-B stress,” at P15, L337 – L340.

In Figure 7d, we show that the genetic interaction between *rpt5a* and *brm* is not additive for the number of meristematic cortex cells under high-B conditions, because we observed an increased number of cells in the *rpt5a brm* double mutant compared with the *rpt5a* single mutant. If it was additive, we would expect to see a further reduction in the number of cells in the *rpt5a brm* double mutant compared with the *rpt5a* single mutant. Therefore, we believe that RPT5a and BRM do not act redundantly.

Comment 4: It has been reported that the yeast bromodomain of SWI/SNF functions to anchor the complex on acetylated nucleosome arrays, leading to retention of the complex on chromatin. Does *Arabidopsis* BRM bind to acetylated histone? Does BRM affect the level of histone acetylation?

Response: Although *Arabidopsis* BRM has been reported to bind histones H3 and H4 (Farrona, et al., 2007), it is still unknown whether acetylated histone residues are required for its binding. However, our analysis of the genome-wide distribution of BRM and acetylated histones using published ChIP-seq data showed the enrichment of BRM around nucleosomes containing acetylated histone residues H3K9ac and H3K14ac. In contrast, BRM was not enriched at H3K27me3 sites as previously reported (Fig. 10b). These data imply that BRM binds to acetylated histone residues. We have added this information to both the results and discussion sections at P20, L465 – L470 and P23 L540 – L545.

To answer your second question, we performed additional experiments to analyze the levels of histone acetylation in the *brm* mutant and transgenic lines with enhanced BRM expression. These experiments showed that BRM does not affect the level of histone acetylation as shown in Fig. 10e. We have added this information to the results at P20, L472 – 473.

Comment 5: Multiple histone modifications could affect chromatin structure. Does high-B stress specifically affect levels of histone acetylation? Does high-B stress influence levels of histone methylation or other modifications?

Response: On the basis of your comment, we performed two additional experiments to analyze the effect of high-B stress on other histone modifications, H3K9me and H3K4me3, which are related to closed and open chromatin, respectively. Total levels of H3K9me were decreased while those of H3K4me3 did not change under high-B stress in both the wild type and *rpt5a* mutants. We have added these data as Supplementary Fig. 11. We could not evaluate the possible changes in the distribution across the genome. These results suggest that high-B stress also affects other histone modifications. However, it is unknown whether the change in other modifications is a consequence of histone hyperacetylation under high-B stress or the direct effects of high-B stress. We have added this explanation at P18, L409 – L420.

Considering a previous report on boric acid as an inhibitor of histone deacetylase *in vitro* (Di Renzo et al., 2007, *Toxicol. Appl. Pharmacol.*) and our results on the relationships between high-B stress and histone acetylation (Fig. 9), high-B stress is inferred to specifically inhibit histone deacetylation.

Comment 6: In Figure5a, it is hard to observe differences of poly-Ub proteins between the wild type and mutants. In this case, it would be more reliable to do replicates and show statistics.

Response: In accordance with the reviewer's suggestion, we have repeated this experiment and added the results of statistical analysis.

Reviewer #3 (Remarks to the Author):

In this work, the authors follow up on previous observations that high B toxicity is manifested through double strand DNA breaks (Plant Cell 25:3533), and provide

a substantial advance in the mechanism through which this occurs.

Specifically:

- They have genetically mapped Arabidopsis boron hypersensitivity mutants (*heb*). They show convincingly that the defects are the result of disruption of specific regulatory subunits of the 19S regulatory particle of the 26S proteasome.
- They also show that only certain paralogs of these proteins affect B sensitivity that argues for subfunctionalization of the regulatory particle subunits.
- By using MS analyses of poly Ub proteins isolated from mutant and wild type plants under high B stress, they have identified 13 proteins that are preferentially represented in the *rpt5a-4* mutant background, and which are presumably targets for proteasome turnover during high B.
- They have focused on one of these, BRM which encodes a chromatin remodeling ATPase. It is shown that BRM over accumulates in *rpt5a-4* plants, and is at least in part responsible for the double stranded breaks and enhanced B toxicity phenotype of these plants.
- Finally, they provide a link between histone hyperacetylation driven by boron (presumably from the inhibition of histone deacetylases), BRM changes in chromatin conformation, which is postulated to lead to increased DNA damage.

This is a substantial body of work, carefully executed, and provides new insights into the molecular basis of boron toxicity through DNA damage. Further, while the work is principally carried out on plants, the authors make a case that this a general mechanism for boron toxicity across species, including humans. However, the authors need to address the following technical and conceptual points:

Response: We appreciate your positive comments that our manuscript provides new insights. We have carefully revised our manuscript to address the technical

and conceptual points you have raised.

Specific comments:

Comment 1: While the authors make a compelling case for a link between BRM accumulation and B toxicity/double stranded breaks, this does not seem to be the complete story. They need to explain the following:

- Wild type plants show B toxicity but show no apparent changes in the level of BRM in response to elevated B (Fig. 6C).

Response: We believe that the level of BRM alone does not correlate with the level of DSBs, as the enhanced expression of BRM alone did not cause increased DSB levels as shown in Supplementary Fig. 10d. We think that BRM enhances genomic instability especially upon increased histone acetylation (Figs. 7–10), which could be a cause of hypersensitivity to high-B stress. In wild type plants, the BRM level is kept low by the 26SP, but histone acetylation is sufficiently elevated to reduce genomic integrity against DNA damage under high-B conditions. Therefore, wild type plants show B toxicity, irrespective of BRM level.

- The authors acknowledge on lines 335-338 that BRM is not the solo cause of hypersensitivity of proteasome mutants to B. (Thus, they should correct the statement in the abstract on lines 36-37 that they “show the molecular basis of high B double strand breaks”). The MS data show that several poly Ub proteins besides BRM are preferentially accumulated in *rpt5a-4* mutants. Some of these could also contribute to B toxicity. Have they been evaluated? At the very least they should be discussed in this context.

Response: Thank you for the comment. We have corrected that sentence in the abstract to “show a molecular pathway of high-B-induced double-strand breaks”.

Among the 12 proteins identified, we have successfully evaluated only TIC20 by generating plants expressing *TIC20* under the control of the 35S promoter. We found that it does not show increased sensitivity to high-B stress

(data not shown). However, we cannot completely exclude the possibility that the other 11 proteins are involved. Thus, we have stated this possibility in the discussion at P24, L563 – P25, L578.

Comment 2: The relationship between high B, histone deacetylase activity, and double stranded breaks is a little bit difficult to follow in the author's narrative. Is the contention that the sole effect of high boron is the inhibition of histone deacetylase activity leading to enhanced chromatin binding? I.e., in this case borate or boric acid has no direct effect on introducing nucleic acid defects? The authors cover this in the discussion but a model would be helpful here. What is the source of DNA damaging/double stranded breaks? Can other effects besides this potential inhibition of Histone deacetylases be at play?

Response: In accordance with your suggestion, we have added a graphical model to our manuscript (Supplementary Fig. 14). In the manuscript, we propose a model for B toxicity; B reduces genomic integrity by inhibiting histone deacetylase activity, which results in high susceptibility to DNA damage of endogenous origin.

The origin of DNA damage under high-B stress has been argued in the academic community but a clear answer has not been given. Thus, we cannot exclude the possibility that B directly causes DSBs; however, this is unlikely considering the chemical properties of borate (it can bind to ribose in RNA but is unlikely to bind DNA; Ricard et al., *Science*, 2004). Therefore, we assume that endogenous factors such as free radical species generated through cellular metabolism are the origins of the DNA breaks. We have added this discussion at P21, L495 – L503.

Comment 3: In their model, the authors should put condensin II into context. In the discussion on pages 21-22, lines 479-484 and SFig. 11, they stated that the double mutant of (*cap-g2-2* and *rpt5a-4*) demonstrate a concerted role in dealing with B genotoxicity. I do not see much difference between single and double

mutants in SFig. 11 and the relationship between condensin and 26SP is not clear.

Response: In Supplementary Fig. 11 (Fig. 15 in the revised manuscript), we predicted that if 26SP and condensin II act in distinct pathways to reduce the toxicity of B, we would see enhanced (additive) high-B sensitivity in a double mutant lacking both functions. However, the results of our additional experiments showed no difference in high-B sensitivity between *rpt5a-4* and the double mutant. Therefore, we conclude that the 26SP and condensin II act in the same pathway. We have discussed this point in more detail at P25, L581 – L584.

In addition, in accordance with the reviewer's suggestion, we put condensin II in our model in the discussion section, P25, L588 – L594.

Comment 4: On lines 376-379, the authors state that “simultaneous treatment of B and TSA resulted in high ratio of root elongation in wild type and *rpt5a-4* compared with single treatment of B” and take this to indicate that B and TSA have a common target. I do not see how this conclusion follows from the data, and the two agents can have distinct or overlapping targets.

Response: In this experiment, we showed that TSA treatment suppresses the inhibitory effects of high-B stress on root elongation as shown by the high ratios of root elongation in the wild type and *rpt5a-4* compared with B treatment alone. One possible explanation for this is the competitive actions of B and TSA. In other words, high-B stress causes histone hyperacetylation. This idea is highly plausible when considering that both TSA and B have been reported as *hdac* inhibitors (Di Renzo et al., 2007, *Toxicol. Appl. Pharmacol.*) and is consistent with the fact that high-B indeed induced histone hyperacetylation in our experiment. If TSA and B have distinct targets, TSA treatment would not affect the sensitivity to high-B stress, or would enhance it. Overall, it is highly plausible that B and TSA have a common target, although we cannot confirm this due to the lack of direct evidence. Therefore, we have weakened our conclusion as follows “high-B stress induces histone hyperacetylation, likely through inhibiting HDAC activity”.

Comment 5: *heb6-1* and *7-1* show substantial inhibition of growth, even under normal B conditions. Are the enhanced B toxicity effects simply due to the fact that these plants are already compromised physiologically?

Response: This is a very important question. To exclude this possibility, we have used the ratio of inhibition to evaluate sensitivity in all experiments. We believe that substantial inhibition of growth is not always linked to stress sensitivity, as some proteasome mutants we tested that had substantial growth inhibition did not show high sensitivity to high-B stress. Even if the enhanced B toxicity is simply due to the plants being physiologically compromised, we can say that a function that complements the growth inhibition affects the sensitivity to high-B stress.

Comment 6:

Lines 254-265 and Figure 5:

- The authors state that there is a higher accumulation of poly-Ub proteins in *rpt5a* but not *rpt5b* mutants. This is not readily apparent from Fig. 5a and there is no quantitation presented.

Response: In accordance with the reviewer's suggestion, we have repeated this analysis and added error bars and statistical analyses.

- Are the proteolytic activities shown in Fig. 5b solely due to 26S protease activities? It seems as if this could be a general proteolytic activity. Can we conclude that these activities are Ub dependent? (as stated in lines 261-2).

Response: When we measured the activities we subtracted the background activities from the total activities to exclude general proteolytic activities caused by factors other than 26S proteasomes. We considered the activities detected upon treatment with proteasome-specific inhibitors as background. This is an

established method and has been reported in several papers (e.g. Kurepa et al., 2008, *Plant J.*; Sakamoto et al., 2011. *Biosci. Biotech. Biochem*; Sun et al., 2013, *J. Proteome Res.*). We have described this method under “Proteasome activity” in the methods section, P31, L725 – P32, L747.

- The authors state that the 26S proteolytic activity is a requirement for high B tolerance. From Fig. 5b it is not clear that 26S proteolytic activity is elevated by high B conditions.

Response: In our manuscript, we claim that 26S proteolytic activity for the degradation of specific targets is required for high-B tolerance. We think that increased 26S activity is not necessary to confer tolerance if the activity is sufficient to degrade the target proteins. When the activity was too low due to impairment caused by factors such as mutations or proteasome inhibitor treatment, the plants showed high sensitivity to high-B stress (Fig. 5b, c). These facts are sufficient to support our conclusion that 26S proteolytic activity is a requirement for high-B tolerance.

Comment 7: It would be appropriate to include more information in the supplementary data regarding the MS data including the numbers of peptides detected, sequence ranges, and the proteins identified in each replicate.

Response: In accordance with the reviewer’s suggestion, we have included the numbers of peptides, scores, and the number of detected proteins for the four replicates for each protein in the MS data file.

Comment 8: For clarity, the authors should indicate the stem cell niche regions in the PI stained micrographs (e.g., Fig. 3a, 7c, 8c). A high magnification micrograph would be informative in terms of showing this region and the effects on meristem cell organization.

Response: In accordance with the reviewer's suggestion, we repeated all microscopic observations and added magnified micrographs to every picture of PI-stained roots.

Comment 9: For Western blot data (e.g., Fig. 6b and c, and Fig. 9 a and h), reproducibility and error bars on histograms are needed. This is especially important since some of the differences (e.g., Fig. 6b and Fig. 9) are not large (less than 2-fold).

Response: In accordance with the reviewer's suggestion, we have repeated these analyses and added error bars and statistical analyses.

Comment 10: On page 17, lines 384 and 385, it is stated that "Histone hyperacetylation is a well-known cause of interphase chromatin decondensation, resulting in impaired chromatin vulnerability...". Do they mean "enhanced" instead of "impaired"?

Response: Thank you for pointing this out; "enhanced" is right. We have corrected this.

Comment 11: Although not a major point of the study, SFig 6 shows that *rpn8a* and *rpn5a* mutants accumulate less B than wild type under elevated B conditions. Are transporters affected?

Response: Thank you for the comment. This would be interesting to see, but it is beyond the scope of our present manuscript to identify a novel mechanism of high-B toxicity. We think that it should be further investigated in future.

Comment 12: Is there a difference in the accumulation of ROS in mutant and

wild type plants under normal or elevated B?

Response: Thank you for the comment. We have already found high ROS accumulation in the root tips under high-B stress. We are about to prepare another manuscript focusing on the relationship between ROS accumulation and high-B stress that will include this result.

Comment 13:

Minor comments:

Fig. 6d is mislabeled.

Fig. 8c has a random “B” that is not defined.

- For sodium butyrate (Fig. 9 and line 507) a different abbreviation besides “NaB” should be used since at first glance this appears to be “boron”.
- Line 394 “were” instead of “was”.
- Where appropriate include v/v or w/v in methods and materials and spell out all abbreviations (e.g., FPS and DMEM on line 591, there are others).

Response: We have corrected all of these points.

Reviewer #4 (Remarks to the Author):

The metalloid element boron (B) is an essential nutrient for plant growth and development. However, in excess it is toxic, generating DNA damage and subsequently resulting in growth inhibition. Here, through a mutagenesis screen for plants being hypersensitive to B, two 26S proteasome (26SP) mutants were identified. Microscopic analysis demonstrated that both genes are essential to maintain root meristem function under high B concentrations. By performing a

differential proteomics screen, the authors identified the BRAMA (BRM) chromatin remodeling complex protein as an 26SP target, and subsequently demonstrated that BRM overproducing and knockout plants display B resistance and sensitivity, respectively. As BRM is known to bind acetylated histone tails, the authors consequently analyzed histone acetylation levels in response to B treatment. Through a set of dedicated experiments, the authors demonstrated that B excess results in an accumulation of histone acetylation (likely through an inhibition of histone deacetylases), resulting in a more open chromatin structure that makes the DNA more accessible to DNA breaking events. It is hypothesized that BRM helps in opening of the chromatin through its chromatin remodeling activity.

This work represents a big step forward in the field. Although high B levels were demonstrated before to trigger a DNA damage response, here the authors convincingly identified the underlying reasons of this toxicity through the generation of a very complete dataset using a number of well-thought-through and clever experiments (e.g. the experiment that helps to discriminate a role for the 26SP in preventing DNA damage). In the end, the data represent an almost completely proven pathway for B toxicity. This excludes direct proof that B inhibits the activity of histone deacetylases, but then again there is sufficient indirect evidence for this supporting this part of the model.

Response: We deeply appreciate your positive comment that our work represents a big step forward in the field. In accordance with your comments, we have made efforts to revise our manuscript to reinforce our proposed model.

Specific comments:

Comment 1: It would help to include a graphical model summarizing all data.

Response: We have added a graphical model to our manuscript in Supplementary Figure 14.

Comment 2: Figure 5a: It would help to include a quantification of the amount of Ub proteins.

Response: In accordance with the reviewer's suggestion, we have repeated this analysis and added error bars and statistical analyses.

Comment 3: Lines 305-308: The authors cannot exclude a difference in translation efficiency in low versus high B, thus delete statement.

Response: Thank you for the comment. We have deleted this statement.

Comment 4: Already in the absence of high B the 26SP mutant displays signs of DNA damage (such as the dead stem cells and upregulation of DNA repair genes). The authors should speculate on the origin of this DNA damage. Is it also linked to more open chromatin structure? Likewise, I would like to see some speculation on by BRM is more targeted for destruction in high versus low B levels.

Response: We believe that increased histone acetylation is a major cause of DNA damage and that BRM abets the genome instability caused by increased histone acetylation. In fact, enhanced BRM expression itself did not cause increased DNA damage but did cause high sensitivity to histone hyperacetylation (Fig. 10d and Supplementary Fig. 10e). In this context, simultaneous high accumulation of histone acetylation and BRM causes genome instability even under normal B conditions, which would be a main reason for the DNA damage in the 26SP mutant. Consistent with this idea, we observed a reduction in a sign of DNA damage, dead stem cells, by impairing the function of BRM in the *rpt5a* mutant under normal B conditions (Fig. 7c). Therefore, we speculate that the targeting of BRM for degradation is similar irrespective of B conditions. We have added this discussion at P23, L552 P24, L562.

Comment 5: Figure 6: Panels are mislabeled.

Response: We have corrected this.

REVIEWERS' COMMENTS:

Reviewer #1 (Remarks to the Author):

Although the revised manuscript is improved, my major concerns were not addressed appropriately. In particular, the molecular mechanism of how 26SP is involved in the tolerance to high B stress is still not defined.

Specific comments:

1. The authors showed that high B stress significantly reduced the number of meristematic cortical cells among the wild type and all high B-hypersensitive RP subunit mutants with much reduced levels in the mutants, indicating that the RP subunits are crucial for the maintenance of RAM organization especially under high B conditions. However, how RP subunits are involved in the maintenance of RAM organization is not clear. Two main pathways, the PLETHORA (PLT) pathway and the SHORT-ROOT (SHR)/SCARECROW (SCR)/RETINOBLASTOMA RELATED pathway act to specify the maintenance of RAM organization. The authors can analyze whether the expression of genes involved these two pathways are affected in RP subunit mutants.

2. The authors claimed that they examined the BRM mRNA levels in the transgenic lines and found a slight correlation between the BRM mRNA level and phenotype. I am not sure what does it means for a slight correlation between the BRM mRNA level and phenotype. Furthermore, the increased BRM expression did not affect the sensitivity to zeocin treatment. Although they claimed that one possible explanation for this phenomenon is a difference in the molecular action leading to DSB accumulation between high-B and zeocin, I think more direct evidence is required to support their hypothesis.

3. Figure 7. Regulation of BRM levels through the 26SP containing RPT5a is crucial for RAM maintenance under high B conditions.

Again, it is important to analyze the genes and pathways that are affected in the mutants responsible for the phenotypes. As indicated by the authors, such work would make it possible to identify the genes regulated by BRM responsible for the maintenance of RAM morphology.

4. Page 16, line 373 -The inhibition of root growth (Figure 8B) and the reduction in RAM size (Figure 8C and 8D) by zeocin treatment were less pronounced in *brm-3* than those in the wild type. Moreover, cell death in the stem cell niche caused by zeocin inducible-DSBs was suppressed in *brm-3* (Figure 8C). These results indicated that BRM has negative effects on the suppression of DSBs caused by various DNA damaging factors, including high B stress.

The molecular mechanisms of why BRM has negative effects on the suppression of DSBs need to be analyzed. Although the authors discussed the possible mechanism in the discussion section, more direct experimental evidence is required.

Reviewer #2 (Remarks to the Author):

The authors have addressed my main concerns.

Reviewer #3 (Remarks to the Author):

In this revised manuscript the authors have addressed (remarkably) the majority of the thirteen

comments brought up in my previous review. The impact of this work in providing important new insights into potential molecular targets of boron toxicity was already apparent in the manuscript that was submitted last fall, and remains in the present manuscript. My remaining comments are aimed at improving the presentation and clarity of the work so that the message does not become lost in the massive amount of data presented (21 figures or supplements, 79 panels!).

1. The working model: I appreciate that the authors have responded to the request to include a unifying model and accompanying that emphasizes the summary of their findings as well as areas that need future attention. However, it is gets lost in the supplementary figures and I would like to see it placed within the main text and discussed in detail. Given the complex array and large quantity of data presented, this is needed to clarify the conclusions for the reader. Specific items needed:

- Potential source of double strand breaks (see below for more detail) associated with B toxicity.
- While emphasizing the clearly demonstrated role of BRM, bring in the possible role of other proteasome targets within the context of the model.
- The authors bring in other chromatin organizational factors such as histone methylation (lines 409-420). How does this fit into the overall model for B toxicity. What needs to be done here in the future.
- At my request, the authors include a discussion of condensing II (Lines 588-94). They should include condensing II in their visual model in Fig. S14.

2. Throughout, the authors refer to "endogenous factors" as the agents triggering the DSB. Please clarify. Do you mean ROS? Are there other agents? Are they enhanced during high B? I realize from their response that they are preparing another paper regarding this (author response to comment 12), but this can be discussed perhaps briefly in the discussion section or within their model.

3. I appreciate the inclusion of higher magnification (comment 8) of root and meristem segments that were assayed for B or other toxin effects. For readers not familiar with these assays, it would be helpful to annotate these images to indicate areas of damage, cell death, or other defects.

4. Pg. 12, lines 279-280. "Our immunoblot analysis showed considerable accumulation of poly-Ub proteins in the roots of rpt5a but not rpt5b-3 mutants.." This seems at odds with this figure which shows strong (the Western is a bit overexposed) accumulation of Ub proteins in all cases. What is meant by "considerable accumulation"?

5. On page 16 line 363-4, the authors conclude "...BRM is not the sole cause of the hypersensitivity of rpt5a mutants to high B stress." They should clarify and elaborate.

6. Similarly, I had difficulty following the author's argument and the relevant details on lines 382-387 regarding how the molecular action leading to DSBs differ between zeocin and high B. Clarification would be helpful.

7. Minor corrections and additions:

- Please indicate in the methods or legends the amount of proteins loaded for immunoblots.
- In abstract, clarify the abbreviation SWI/SNF.
- Line 291 remove "itself".

- The term “overfunction” on line 349 is vague, what is meant by this?
- Figure 7e is vague, it is not clear which data sets were used to generate this figure.
- The y-axis label on Fig. 7b “ $1/0.03\text{mM B}$ ” is strange and seems to indicate the reciprocal of the B concentration.
- Supplementary Fig. 10, panels c and d are reversed.

Point-by-point Responses to the Reviewers' Comments

We appreciate the valuable comments from two reviewers. We have carefully revised the manuscript in accordance with their comments and suggestions. The revised parts are shown by track changes in the revised manuscript, and detailed point-by-point responses are provided below.

Reviewer #1 (Remarks to the Author):

Although the revised manuscript is improved, my major concerns were not addressed appropriately. In particular, the molecular mechanism of how 26SP is involved in the tolerance to high-B stress is still not defined.

Response: Thank you for evaluating our revised manuscript. We believe that our experiments could clearly address a molecular mechanism of how 26SP involved in the tolerance to high B stress: 26SP prevents negative effects of BRM that causes DSBs by degrading BRM protein. As you mentioned, actually, the molecular action of BRM causing DSBs still needs to be analyzed. Additionally, as shown by our proteome analysis, there might be some other candidates involved in the tolerance to high-B stress, suggesting possibility of other molecular mechanisms to be analyzed. We also consider that unraveling of these things is required to fully understand the mechanism of high-B stress. However, these works are beyond the scope of our present manuscript. As a next step in the future, we would like to design a new project based on your constructive suggestions.

Specific comments:

Comment 1: The authors showed that high B stress significantly reduced the number of meristematic cortical cells among the wild type and all high B-hypersensitive RP subunit mutants with much reduced levels in the mutants, indicating that the RP subunits are crucial for the maintenance of RAM organization especially under high B conditions. However, how RP subunits are involved in the maintenance of RAM organization is not clear. Two main

pathways, the PLETHORA (PLT) pathway and the SHORT-ROOT (SHR)/SCARECROW (SCR)/RETINOBLASTOMA RELATED pathway act to specify the maintenance of RAM organization. The authors can analyze whether the expression of genes involved these two pathways are affected in RP subunit mutants.

Response: The RAM organization in RP mutants treated with high-B was severely defected (some of them lose meristem). Therefore it is assumed that both two main pathways are affected. In addition, it is difficult to define whether such changes in gene expression are due to the direct or indirect consequence of 26SP dysfunction. Therefore, even if we analyze the expression levels of your suggested genes, it seems not to give us new insights into the molecular mechanism of how 26SP maintains RAM under high-B stress. Thus, we do not include the expression analysis of genes involved in RAM maintenance in the revised manuscript. As shown in Figure 7, we could demonstrate that the defects in RAM organization of RP mutants are largely dependent on the BRM level. In line with the results shown in Figure 6, we could conclude that the regulation of BRM by RP is important for the RAM maintenance. Therefore, we believe that “how RP subunits are involved in the maintenance of RAM organization” is already addressed in our manuscript.

Comment 2: The authors claimed that they examined the BRM mRNA levels in the transgenic lines and found a slight correlation between the BRM mRNA level and phenotype. I am not sure what does it means for a slight correlation between the BRM mRNA level and phenotype. Furthermore, the increased BRM expression did not affect the sensitivity to zeocin treatment. Although they claimed that one possible explanation for this phenomenon is a difference in the molecular action leading to DSB accumulation between high-B and zeocin, I think more direct evidence is required to support their hypothesis.

Response: As you mentioned, “a slight correlation” is not a proper representation. As we could see the positive relationship between mRNA levels

and phenotype, we would like to say that there is a positive correlation.

As we showed in Figure 10, the exacerbation of negative effect of BRM is caused by histone hyperacetylation. Therefore, it is conceivable that the difference in sensitivity of enhanced BRM expression lines between high-B and zeocin treatment is attributed to the capability of inducing histone hyperacetylation. Because zeocin is a radiomimetic drug, it does not induce histone hyperacetylation. This fact supports our idea regarding to the difference. We added this discussion at P20, L490-P21, L494.

Comment 3: Figure 7. Regulation of BRM levels through the 26SP containing RPT5a is crucial for RAM maintenance under high B conditions.

Again, it is important to analyze the genes and pathways that are affected in the mutants responsible for the phenotypes. As indicated by the authors, such work would make it possible to identify the genes regulated by BRM responsible for the maintenance of RAM morphology.

Response: Thank you for the constructive comment. Actually, gene expression analysis may give us some clues for the downstream actions of BRM involved in the maintenance of RAM morphology. However, it is considered that gene expression analysis alone is not sufficient to identify such responsible genes. We would like to identify such genes through performing several kinds of experiments in our future project.

Comment 4: Page 16, line 373 -The inhibition of root growth (Figure 8B) and the reduction in RAM size (Figure 8C and 8D) by zeocin treatment were less pronounced in *brm-3* than those in the wild type. Moreover, cell death in the stem cell niche caused by zeocin inducible-DSBs was suppressed in *brm-3* (Figure 8C). These results indicated that BRM has negative effects on the suppression of DSBs caused by various DNA damaging factors, including high B stress.

The molecular mechanisms of why BRM has negative effects on the suppression of DSBs need to be analyzed. Although the authors discussed the possible mechanism in the discussion section, more direct experimental evidence is required.

Response: Thank you for pointing it out. We are also so much interested in the molecular mechanism of why BRM has negative effects on the suppression of DSBs. Considering from our results, it is likely that the binding of BRM to acetylated histone has a key process to evoke genomic instability. We would like to fully unveil this remaining issue in the future work.

Reviewer #2 (Remarks to the Author):

The authors have addressed my main concerns.

Response: Thank you very much for evaluating our manuscript.

Reviewer #3 (Remarks to the Author):

In this revised manuscript the authors have addressed (remarkably) the majority of the thirteen comments brought up in my previous review. The impact of this work in providing important new insights into potential molecular targets of boron toxicity was already apparent in the manuscript that was submitted last fall, and remains in the present manuscript. My remaining comments are aimed at improving the presentation and clarity of the work so that the message does not become lost in the massive amount of data presented (21 figures or supplements, 79 panels!).

Comment 1: The working model: I appreciate that the authors have responded to the request to include a unifying model and accompanying that emphasizes the summary of their findings as well as areas that need future attention. However, it is gets lost in the supplementary figures and I would like to see it placed within the main text and discussed in detail. Given the complex array and

large quantity of data presented, this is needed to clarify the conclusions for the reader. Specific items needed:

- Potential source of double strand breaks (see below for more detail) associated with B toxicity.
- While emphasizing the clearly demonstrated role of BRM, bring in the possible role of other proteasome targets within the context of the model.
- The authors bring in other chromatin organizational factors such as histone methylation (lines 409-420). How does this fit into the overall model for B toxicity. What needs to be done here in the future.
- At my request, the authors include a discussion of condensin II (Lines 588-94). They should include condensin II in their visual model in Fig. S14.

Response: Thank you very much for the constructive comments to improve our working model. We tried to transfer the model to main figure, but we could not achieve it due to the limitation of number of displays in main text. According to your comments, we added the description of potential source of DBS induction and possibility of histone hyperacetylation by other proteasomal targets in our model.

As to reduction of histone methylation, it may be just a consequence of epigenetic crosstalk. Or certain targets of proteasome may mediate the change in histone methylation. Or boron may act as histone methyltransferase inhibitor. However, at this moment, we do not have any clues for confirming these ideas. Therefore it is difficult to bring this modification in our model. We would like to evaluate the relationship between boron action and histone methylation in the future.

Similarly, the discussion about the functional relationship between condensin II and proteasome is merely a speculation. We concern that readers misunderstand that we also found a molecular mechanism of condensin II. We would like to make a more solid model including condensin II by obtaining

reliable data in the future work.

Comment 2: Throughout, the authors refer to “endogenous factors” as the agents triggering the DSB. Please clarify. Do you mean ROS? Are there other agents? Are they enhanced during high B? I realize from their response that they are preparing another paper regarding this (author response to comment 12), but this can be discussed perhaps briefly in the discussion section or within their model.

Response: According to your comment, we discussed about the possible endogenous factors that triggers DSBs at P21, L510-P22, 524. At this moment, we cannot conclude that ROS is the main endogenous factor. In our manuscript, “endogenous factors” mean hydrolysis, ROS and other reactive metabolites.

Comment 3: I appreciate the inclusion of higher magnification (comment 8) of root and meristem segments that were assayed for B or other toxin effects. For readers not familiar with these assays, it would be helpful to annotate these images to indicate areas of damage, cell death, or other defects.

Response: According to your comment, we added the description for magnified images with the figure legend.

Comment 4: Pg. 12, lines 279-280. “Our immunoblot analysis showed considerable accumulation of poly-Ub proteins in the roots of rpt5a but not rpt5b-3 mutants”. This seems at odds with this figure which shows strong (the Western is a bit overexposed) accumulation of Ub proteins in all cases. What is meant by “considerable accumulation”?

Response: Thank you for the comment. As you pointed out, “considerable accumulation” is not a suitable depiction. In the revised manuscript, we changed

this to “Our immunoblot analysis showed *increased* accumulation of poly-Ub proteins in the roots of *rpt5a* but not *rpt5b-3* mutants *compared with the wild type*”.

Comment 5: On page 16 line 363-4, the authors conclude “...BRM is not the sole cause of the hypersensitivity of *rpt5a* mutants to high B stress.” They should clarify and elaborate.

Response: As you mentioned, this conclusion is not clear. Although the defects of BRM in *rpt5a* mutant largely rescued its root growth, we can still see some defects in RAM. Therefore we considered that there might be some other factors involved in the hypersensitivity of the *rpt5a* mutants to high-B stress, in addition to the adverse effect on root growth caused by the overaccumulation of BRM. We changed the conclusion to “the overaccumulation of BRM is not the sole cause of the hypersensitivity of *rpt5a* mutants to high-B stress”.

Comment 6: Similarly, I had difficulty following the author’s argument and the relevant details on lines 382-387 regarding how the molecular action leading to DSBs differ between zeocin and high B. Clarification would be helpful.

Response: We found that BRM increases the negative effects of histone hyperacetylation on genomic stability (Figure 10). In other words, histone hyperacetylation is required for the action of BRM leading to DSBs induction. Therefore, we consider that the difference is mainly attributed to the capability of histone hyperacetylation. Considering that zeocin is a radiomimetic drug that can cause oxidative damage in DNA through producing ROS, it does not induce histone hyperacetylation. We added this discussion at P20, L490-P21, L494.

Minor corrections and additions:

Comment 1: Please indicate in the methods or legends the amount of proteins

loaded for immunoblots.

Response: According to your comment, we added the information on amount of proteins in the methods section.

Comment 2: In abstract, clarify the abbreviation SWI/SNF.

Response: According to your comment, we clarified the abbreviation “SWI/SNF” in both abstract and introduction.

Comment 3 Line 291 remove “itself”.

Response: According to your comment, we removed “itself”.

Comment 4: The term “overfunction” on line 349 is vague, what is meant by this?

Response: Here, we would like to question whether increased accumulation of BRM has negative function that evokes reduced tolerance to high-B stress. We expressed such function by using a word “overfunction”. But according to your indication, it does not seem to be a proper expression. Therefore, we changed the term “overfunction” to “overaccumulation”.

Comment 5: Figure 7e is vague, it is not clear which data sets were used to generate this figure.

Response: To generate this figure, we used same imaging data sets of root meristem treated with 3 mM B used for Figure 7d. Thank you for pointing it out, as we noticed that the number of “n” is wrong. We corrected the number of “n”

and added the description which data sets were used in the legend.

Comment 6: The y-axis label on Fig. 7b “/ 0.03mM B” is strange and seems to indicate the reciprocal of the B concentration.

Response: According to your comment, we deleted “/ 0.03 mM B” from the y-axis label and changed the explanatory notes of the figure to “1.5 mM B/ 0.03 mM B” and “3 mM B/ 0.03 mM B”.

Comment 7: Supplementary Fig. 10, panels c and d are reversed.

Response: According to your comment, we replaced the position of panel c with that of panel d.